



# Controls on Early Cretaceous South Atlantic Ocean circulation and carbon burial - a climate model-proxy synthesis

Sebastian Steinig[1,6], Wolf Dummann[2,7], Peter Hofmann[2], Martin Frank[1], Wonsun Park[1,3,4], Thomas Wagner[5], and Sascha Flögel[1]

[1]GEOMAR Helmholtz Centre for Ocean Research Kiel, Kiel, Germany
[2]Institute of Geology and Mineralogy, University of Cologne, Cologne, Germany
[3]Center for Climate Physics, Institute for Basic Science (IBS), Busan, Republic of Korea
[4]Department of Climate System, Pusan National University, Busan, Republic of Korea
[5]The Lyell Centre, Heriot–Watt University, Edinburgh, UK
[6]Present address: School of Geographical Sciences, University of Bristol, Bristol, UK
[7]Present address: Institute of Geosciences, Goethe-University Frankfurt, Frankfurt am Main, Germany

**Correspondence:** Sebastian Steinig (sebastian.steinig@bristol.ac.uk)

**Abstract.** Black shale sediments from the Barremian to Aptian South Atlantic document intense and widespread burial of marine organic carbon during the initial stages of seafloor spreading between Africa and South America. The enhanced sequestration of atmospheric $CO_2$ makes these young ocean basins potential drivers of the Early Cretaceous carbon cycle and climate perturbations. The opening of marine gateways between initially restricted basins and related circulation and ventilation changes are a commonly invoked explanation for the transient formation and disappearance of these regional carbon sinks. However, large uncertainties in paleogeographic reconstructions limit the interpretation of available paleoceanographic data and prevent any robust model-based quantifications of the proposed circulation and carbon burial changes. Here, we present a new approach to assess the principal controls on the Early Cretaceous South Atlantic and Southern Ocean circulation changes under full consideration of the uncertainties in available boundary conditions. Specifically, we use a large ensemble of 36 climate model experiments to simulate the Barremian to Albian progressive opening of the Falkland Plateau and Georgia Basin gateways with different configurations of the proto-Drake Passage, the Walvis Ridge, and atmospheric $CO_2$ concentrations. The experiments are designed to complement available geochemical data across the regions and to test circulation scenarios derived from them. All simulations show increased evaporation and intermediate water formation at subtropical latitudes that drive a meridional overturning circulation whose vertical extent is determined by the sill depth of the Falkland Plateau. Densest water masses formed in the southern Angola Basin and potentially reached the deep Cape Basin as Walvis Ridge Overflow Water. Paleogeographic uncertainties are as important as the lack of precise knowledge of atmospheric $CO_2$ levels for the simulated temperature and salinity spread in large parts of the South Atlantic. Overall temperature uncertainties are up to 15 °C and increase significantly with water depth. The ensemble approach reveals temporal changes in the relative importance of geographic and radiative forcings for the simulated oceanographic conditions and, importantly, nonlinear interactions between them. Progressive northward opening of the highly restricted Angola Basin increased the sensitivity of local overturning and upper ocean stratification to atmospheric $CO_2$ concentrations due to large-scale changes in the hydrological cycle, while the chosen proto-Drake Passage depth is critical for the ocean dynamics and $CO_2$ response in the southern South Atlantic. Finally,



the simulated processes are integrated into a recent carbon burial framework to document the principal control of the regional gateway evolution on the progressive shift from the prevailing saline and oxygen-depleted subtropical water masses to the dominance of ventilated high-latitude deep waters.

## 1 Introduction

The Early Cretaceous (i.e., Barremian to Albian) South Atlantic Ocean provides a unique possibility to gain insights into the complex interplay of large-scale tectonics, related ocean circulation and carbon burial changes and the global climate evolution. On the one hand, the young ocean basin emerged in the wake of the ongoing breakup of Gondwana and the rifting between South America and Africa that had started during the earliest Cretaceous (Torsvik et al., 2009; Heine et al., 2013). On the other hand, the general warmth of the Early Cretaceous greenhouse climate (O'Brien et al., 2017) was interrupted by severe perturbations of the global carbon cycle (e.g., Weissert, 1989; Bralower et al., 1994), short-term temperature fluctuations (Bodin et al., 2015; Jenkyns, 2018; Cavalheiro et al., 2021) and marine biotic crises (McAnena et al., 2013). Large shelf areas and restricted circulation in young ocean basins produced favourable conditions for increased biogeochemical turnover rates and organic carbon burial (Trabucho-Alexandre et al., 2012), thus providing a possible link between regional basin evolution and global carbon cycle perturbations. Sedimentary evidence for periods of enhanced carbon sequestration come from the widespread deposition of organic-rich black shales found throughout the Mesozoic Atlantic Ocean, with an increased occurrence during the Aptian-Albian (Stein et al., 1986). Biogeochemical modelling showed that the restricted environments of the developing South Atlantic and Southern Ocean acted as efficient carbon sinks that played a crucial role in driving global cooling during the Late Aptian (McAnena et al., 2013).

Dummann et al. (2020, 2021a) reconstructed the history of Early Cretaceous gateway openings in the southern South Atlantic and the related formation and disappearance of regional carbon sinks. They combined a new stratigraphic framework for distinct sites across the South Atlantic and Southern Ocean with radiogenic neodymium (Nd) isotope signatures to trace past water mass mixing between both basins (Fig. 1b). The neodymium isotopic composition, expressed as $\epsilon_{Nd}(t)$, is used as a quasi-conservative tracer of bottom water masses (Frank, 2002). Differences in the $\epsilon_{Nd}(t)$ of seawater are mainly caused by spatially heterogeneous lithogenic input in the source region of a water mass and by mixing along its flow path. Dummann et al. (2020) demonstrated the dominant control of marine gateways on the local ocean circulation and ventilation. Prior to 117 Ma, the paleogeographic position of the Falkland Plateau severely limited water mass exchange between the South Atlantic and the Southern Ocean (Fig. 1c). Limited ventilation of the Cape Basin led to oxygen-deficient deep waters and formation of organic-rich black shales (Fig. 1a). The detection of a more radiogenic water mass at DSDP Site 511 on the Falkland Plateau at around 117 Ma marks the onset of increased South Atlantic Intermediate Water (SAIW) export due to the progressive westward drift of the Falkland Plateau (Fig. 1d). Between 113 and 110 Ma, the Maurice Ewing Bank cleared the southern tip of Africa and allowed the first deep water exchange between the South Atlantic and Southern Ocean via the Georgia Basin Gateway (Fig. 1e). Less radiogenic $\epsilon_{Nd}(t)$ signatures in the South Atlantic indicate stronger advection of cold oxygenated waters from the Southern Ocean. As a consequence, the enhanced organic carbon burial ended even in the deep South Atlantic.



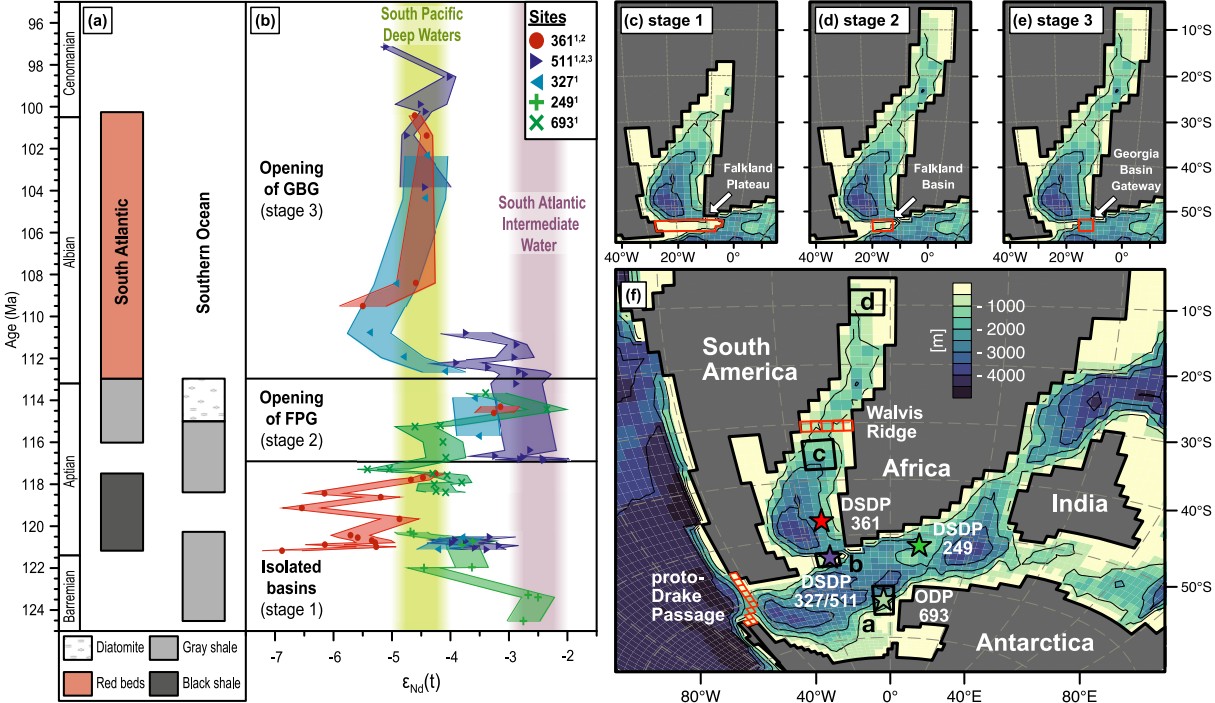

**Figure 1.** Reconstructed water mass evolution and model set-up. Panels (a-b) are modified from Dummann et al. (2020) and updated to GTS2020 and show the (a) dominant lithology of sites in the South Atlantic and Southern Ocean and (b) evolution of seawater Nd isotope signatures with ±2 SD range. Green and purple shading indicates the proposed Nd isotope range of the intermediate and deep water end members of the South Pacific and South Atlantic. Literature $\epsilon_{Nd}(t)$ values are taken from Dummann et al. (ref[1], 2020), Murphy and Thomas (ref[2], 2013) and Robinson et al. (ref[3], 2010). Panels (c-e) show applied model bathymetries for stages 1 to 3 and (f) the approximate paleolocations of the study sites adapted to the stage 2 model bathymetry. Highlighted grid points for the proto-Drake Passage and Walvis Ridge show maximum applied gateway depth. Black rectangles in (f) represent areas used for averaging in Fig. 6.

Climate models are an ideal tool for testing these proxy-derived hypotheses and quantifying the associated regional circulation and paleoenvironmental changes. Previous modelling studies support the idea that large-scale ocean circulation in the young South Atlantic and Southern Ocean was primarily controlled by the paleogeography (Chaboureau et al., 2012; Uenzelmann-Neben et al., 2017; Ladant et al., 2020). But this also means that the validity of any simulated mean state and potentially also the predicted response to the proposed gateway changes critically depend on the quality of applied model boundary conditions. While this poses a problem to paleomodelling applications in general, it becomes even more important for analyses of regional scales for deep-time periods. For the Early Cretaceous South Atlantic opening, we identify 3 major uncertainties in available boundary conditions:

1. Stomata-based reconstructions of Barremian-Albian atmospheric $CO_2$ concentrations vary between 400-1500 ppmv (Wang et al., 2014; Jing and Bainian, 2018).





2. Estimates of the proto-Drake Passage (DP, Fig. 1f) bathymetry for the Early Cretaceous range from intermediate depth (Sewall et al., 2007; Eagles, 2010) to shallow marine (Cao et al., 2017) to a complete closure (Hay et al., 1999).

3. The depth of the Walvis Ridge (WR), which separates the southern South Atlantic from the Angola Basin (Fig. 1f), is poorly constrained (Pérez-Díaz and Eagles, 2017).

All of these uncertainties have the potential to influence the oceanographic response to changes in the paleogeographic position of the Falkland Plateau. Traditionally, dedicated sensitivity experiments are used to diagnose the influence of each uncertain parameter on the simulated circulation individually (e.g., Barron and Washington, 1984; Sloan and Rea, 1996; Chaboureau et al., 2012; Ladant et al., 2020). Although computationally efficient, this approach does not necessarily capture all possible circulation states, because it does not account for potential nonlinear responses. For example, the simulated response to changes
in atmospheric $CO_2$ concentrations could be different during individual opening stages of the South Atlantic or might itself depend on the chosen depth of the Drake Passage or Walvis Ridge. To resolve this dilemma, we propose an ensemble approach to simulate the first three stages of the proposed gateway history under full consideration of the above-mentioned uncertainties. We adapt a concept similar to ensemble forecasting used for numerical weather prediction, but instead of accounting for imperfect initial conditions, we account for uncertainties in the boundary conditions. We make use of advances in available
computing power to generate a large ensemble of 36 model simulations (3 opening stages $\times$ 2 $CO_2$ levels $\times$ 3 DP depths $\times$ 2 WR depths) instead of dedicated sensitivity experiments. This approach allows us to estimate the full range of possible model results arising from the uncertainties in the boundary conditions and to identify potential nonlinear responses of the regional ocean circulation through time. We further use the spread between different ensemble members to assess the robustness of our results and derived conclusions. Finally, we constrain the range of simulated circulation scenarios with the reconstructed water
mass mixing to derive an integrated history of ocean circulation and carbon burial for the Early Cretaceous South Atlantic and Southern Ocean.

## 2   Methods

### 2.1   Model description

We use a version of the Kiel Climate Model (KCM, Park et al., 2009), a fully coupled atmosphere-ocean-sea ice general
circulation model, for all simulations. It uses the ECHAM5 spectral atmospheric general circulation model (Roeckner et al., 2003), with 19 vertical levels and a higher horizontal resolution of $\sim$2.8$°$ x 2.8$°$ (T42) than described in Park and Latif (2019). It is coupled to the NEMO ocean-sea ice model (Madec, 2008), which uses a tripolar grid with a horizontal resolution of 2$°$ (ORCA2) at 31 vertical levels with an equatorial refinement of the meridional resolution to 0.5$°$. The global land topography and ocean bathymetry (Blöhdorn, 2013) represent an Early Aptian ($\sim$120 Ma) time slice and are based on reconstructions
of Müller et al. (2008) and Blakey (2008). Freshwater routing from the land surface to the ocean follows the land orography (Hagemann and Dümenil, 1998). Simplified climate-zone-dependent vegetation parameters (Blöhdorn, 2013) are applied to the ice-free land surface. Integrations are performed at atmospheric $CO_2$ concentrations of 600 and 1200 ppmv to account for



the large spread in available reconstructions for the Early Cretaceous (e.g., Wang et al., 2014; Jing and Bainian, 2018). The solar constant is reduced by ∼1% to 1350 W/m$^2$, while the orbital parameters and other greenhouse gases are fixed at their

preindustrial values due to the lack of available reconstructions.

## 2.2 South Atlantic and Southern Ocean paleobathymetry

The model setup is nearly identical to the simulations in Steinig et al. (2020). But to increase comparability with previous studies (Chaboureau et al., 2012; Uenzelmann-Neben et al., 2017) and improve reproducibility, we replaced the regional ocean bathymetry of the study area with published climate model boundary conditions for the Early Cretaceous (Sewall et al., 2007).

To focus on gateway-related circulation changes during the initial stages of the South Atlantic opening, we kept the large-scale bathymetry of the Southern Ocean and southern South Atlantic fixed throughout all simulations. Sewall et al. (2007) provide paleobathymetries for the time slices of the Early Aptian and Early Albian. We chose the latter as our base configuration (Fig. 1f) for two reasons. First, it features an initial shallow-water connection between India and Antarctica. This proto-Indian Ocean is missing in the Early Aptian time slice, but is present in other reconstructions (Golonka, 2007; Gibbons et al., 2013) and other

climate model boundary conditions for the earliest Cretaceous (Lunt et al., 2016). Second, it includes a shallower maximum water depth of the DP of ∼1400 m compared to ∼2000 m in the Early Aptian time slice (Sewall et al., 2007). This is more in line with the general view of a shallow or intermediate water connection between the Pacific Ocean and the Weddell Sea during the Cretaceous (e.g., Eagles, 2010; Donnadieu et al., 2016; Cao et al., 2017), while the onset of deep water exchange was associated with the formation of oceanic crust starting in the Oligocene (Lagabrielle et al., 2009). To further account

for the considerable uncertainty in estimates of the DP depth, ranging from about 1400 m (Sewall et al., 2007) to shallow marine (Cao et al., 2017) to a closed gateway (Hay et al., 1999), we perform the simulations with three different depths of the passage of 1400 m, 200 m and 0 m, respectively (200 m and 0 m depths are applied to all highlighted DP grid points in Fig. 1f). Paleobathymetric reconstructions for the WR between the Cape-Argentine and Angola basins (Fig. 1f), are similarly uncertain (Pérez-Díaz and Eagles, 2017). Although previous modelling studies assumed relatively large sill depths of ∼1000

m (Lunt et al., 2016; Uenzelmann-Neben et al., 2017), plate tectonic modelling (Heine et al., 2013) and seawater Nd-isotope data (Dummann et al., 2023) suggest little to no water mass exchange across the WR during the Aptian-Albian. We therefore conducted two sets of simulations with a maximum depth of the WR of either ∼1200 m (Sewall et al., 2007) or 200 m (a depth of 200 m is applied to all highlighted WR grid points in Fig. 1f). In contrast to the DP, we do not consider a completely closed WR due to evidence of at least some connectivity between the southern and northern South Atlantic (e.g. Dingle, 1999;

Kochhann et al., 2014; Cui et al., 2023).

Due to the proposed crucial role of the Falkland Plateau in governing the water mass exchange between the South Atlantic and the Southern Ocean, we refined its representation in the local model bathymetry. Shape and position are based on a high-resolution regional study (see Supplementary Fig. S1, Pérez-Díaz and Eagles, 2017), but the limited model resolution only allows for an idealised geometry (red polygon in Fig. 1c). The plateau consists of a shallow (250 m) Malvinas Basin in the

West, a deeper Falkland Basin (750 m, Fig. 1d) and a shallow (250 m) Maurice Ewing Bank in the East. The chosen depth of the Falkland basin is in between a reconstructed Aptian shelf environment (Harris et al., 1977; Basov and Krasheninnikov,



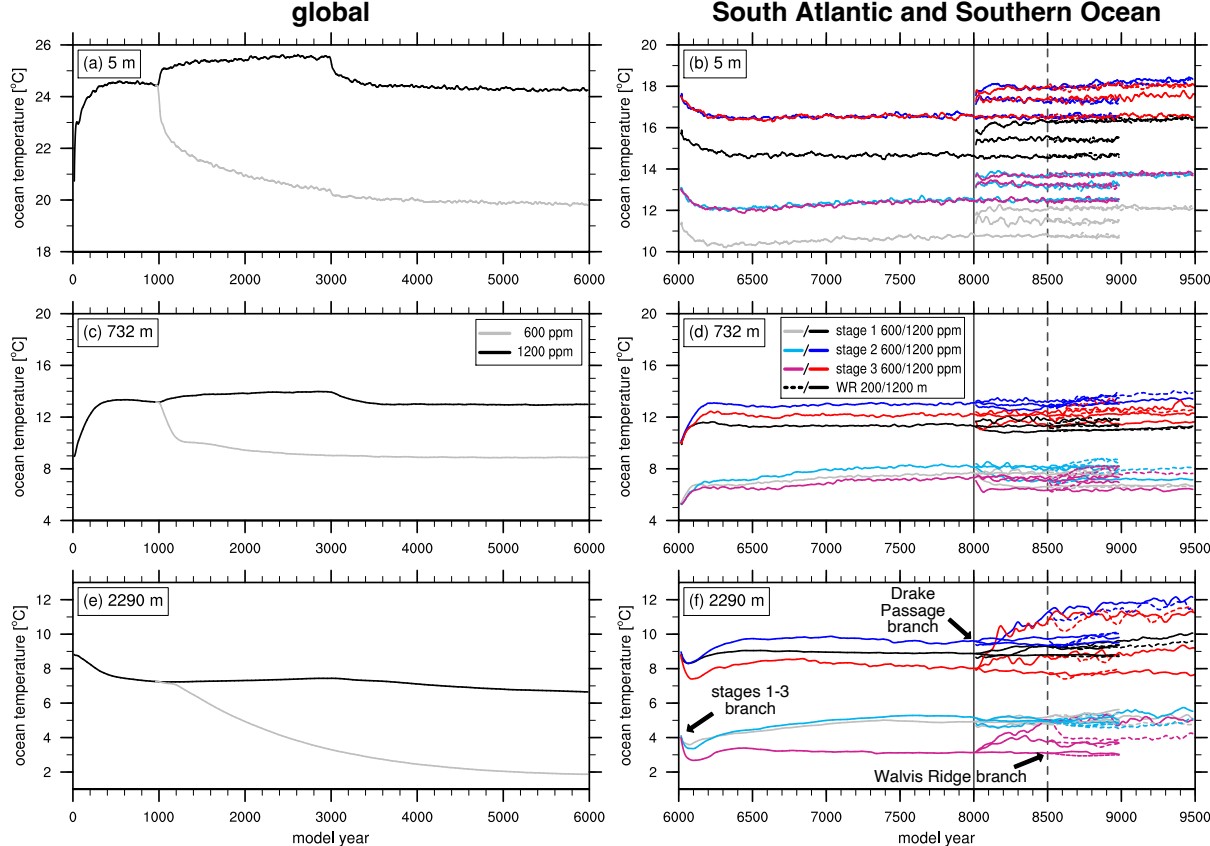

**Figure 2.** Overview of ensemble spin-up phases. Time series of annual mean ocean temperatures at different depths are averaged globally on the left and over the South Atlantic and Southern Ocean (0°-90°S, 40°W-20°E) on the right. Color coding represents the different stages defined in the main text and the two levels of atmospheric $CO_2$. The sudden shifts in temperature at years 1000 and 3000 of the initial spin-up are related to a temporary increase of the atmospheric time step length, which has been reversed to improve model stability.

1983) and a Late Albian water depth of ∼800 m (Holbourn et al., 2001). The estimated subsidence based on assemblages of benthic foraminifera was supported by the application of backtracking methods (Müller et al., 2018). The depth of the Georgia Basin Gateway (Fig. 1e) was set to 2500 m (Pérez-Díaz and Eagles, 2017).

## 2.3 Ensemble approach and model spin-up

We define model geographies for three consecutive time slices representing the initial stages of South Atlantic opening (Fig. 1c-e). They resemble the proposed characteristics of the first three stages of the reconstructed water mass exchange between the South Atlantic and Southern Ocean (see Section 1) associated with a progressive westward drift of the Falkland Plateau. The bathymetry of the Southern Ocean and southern South Atlantic (i.e. the Cape-Argentine Basin) is fixed for all stages due to the large uncertainties in the available reconstructions (Pérez-Díaz and Eagles, 2017). In fact, at the upper end of reconstructions,



the southern South Atlantic at 120 Ma is as wide and even deeper than the minimum estimate at 110 Ma (see Supplementary Fig. S1). For the region north of the WR we apply the Early Aptian bathymetry of Sewall et al. (2007) for stage 1 and the Early Albian time slice for stages 2 and 3 to assess the influence of the changing Angola Basin geometry. As the applied topography around the Angola Basin is fixed to an Early Aptian time slice and does not include the Early Albian equatorial rift valley

shown in Sewall et al. (2007), we add a drainage basin north of the Angola Basin to the model freshwater routing for stages 2 and 3. These three opening stages are simulated with all possible combinations of sensitivity parameters discussed above, i.e. variable DP depth (0/200/1400 m), variable WR depth (200/1200 m) and two atmospheric $CO_2$ end members (600/1200 ppmv). This results in 12 different ensemble members for each stage and 36 simulations in total.

Throughout the analysis, we will focus on the oceanographic changes associated with four key processes defined as:

1. a doubling of atmospheric $CO_2$, i.e. 1200 minus 600 ppmv (hereafter $\Delta CO_2$)

2. an opening of the proto-Drake Passage, i.e. original maximum depth of 1400 m minus closed gateway (hereafter $\Delta DP$)

3. an opening of a deep water connection between the South Atlantic and Southern Ocean via the Georgia Basin, i.e. stage 3 minus stage 2 (hereafter $\Delta GBG$)

4. a deeper Walvis Ridge, i.e. 1200 m minus 200 m (hereafter $\Delta WR$)

If not stated differently, these changes are averaged over all respective ensemble members, i.e. results for $\Delta CO_2$ represent the average difference between 18 simulations with 1200 ppmv and 18 simulations with 600 ppmv $CO_2$. All ensemble members are equally weighted for the calculation of the ensemble averages.

Simulations are initialised with the output of two existing 6000-year-long integrations (Steinig et al., 2020) using the boundary
conditions described in Section 2.1 at 600 and 1200 ppmv $CO_2$, respectively. These two simulations were started from a homogeneous ocean with a temperature of 10 °C and a salinity of 35. The remaining temperature drift even in the deeper ocean was below 1 °C over the last 1000 years for both spin-ups (Fig. 2e). From this, the six experiments for stages 1 to 3 at the two chosen $CO_2$ levels were branched off and integrated for another 2000 years to allow the regional circulation to adapt to the modified geography described in Section 2.2. Ensemble simulations with changes in the depth of the DP (WR) are branched
off at overall model year 8000 (8500) and integrated for a further 1000 (500) years. Ensemble members with a closed DP are integrated for an additional 500 years to account for the larger regional temperature response. Results are averaged over the respective last 100 model years at which the simulations approach a quasi-equilibrium even for the deep South Atlantic and Southern Ocean (Fig. 2f).



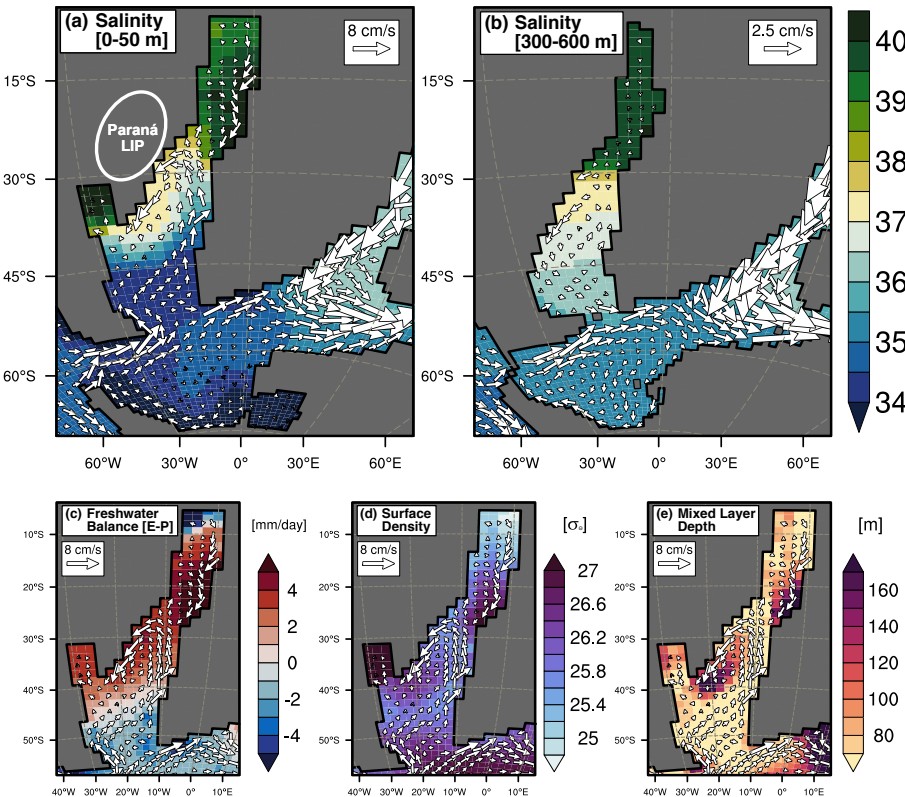

**Figure 3.** Simulated mean ocean circulation averaged over all 36 ensemble simulations. Only grid points that represent ocean values in at least half of the simulations are shown. Panels show annual mean ocean velocity and salinity averaged over (a) 0-50 m and (b) 300-600 m, as well as 0-50 m velocity on top of the (c) net freshwater flux from the ocean to the atmosphere, (d) sea surface density, and (e) climatological maximum monthly mean mixed layer depth. Velocity vector length is limited to two times the reference magnitude for figure clarity. White oval in (a) represents approximate position of the Paraná Large Igneous Province (LIP).

## 3 Results

### 3.1 Ensemble mean circulation

The ensemble mean fields, i.e. the averages across all 36 simulations, provide a robust estimate of the main circulation features and are used as a baseline to further test the sensitivity to individual boundary condition changes. The Southern Ocean circulation is dominated by the prevailing westerly winds that drive surface water inflow from the South Pacific in case of an open DP (Fig. 3a). After passing the southern tip of South America, a surface current of ∼5 cm/s enters the southern South Atlantic via the Malvinas Basin and carries relatively fresh southern-sourced surface waters along the eastern boundary northwards. Surface circulation in the narrow South Atlantic is cyclonic in the Angola Basin and anticyclonic south of the WR. Surface salinities are enhanced in the southward return flows of these small-scale gyres due to enhanced rates of evaporation at sub-





tropical latitudes (Fig. 3c). The associated higher surface densities (Fig. 3d) enhance vertical mixing (Fig. 3e) and lead to the formation of South Atlantic Intermediate Water (SAIW) offshore South America at ∼35°S. As the WR severely limits lateral

water mass exchange, the deeper Angola Basin is characterised by a sluggish circulation and is filled with a locally produced saline water mass (Fig. 3b). SAIW formed in the vicinity of the Paraná large igneous province (Fig. 3a) exits the South Atlantic via the Falkland Plateau at flow speeds of 1-2 cm/s, mixes with Southern Ocean waters and eventually merges with the southern limb of the strong subtropical gyre circulating around India (Fig. 3b).

## 3.2 Temperature and salinity

As a result of the salinity-driven intermediate water formation, the southern South Atlantic and the Angola Basin both show pronounced subsurface maxima in temperature and salinity at depths between 200-600 m (Fig. 4a-b). The deeper parts of the silled basins are filled with relatively warm and saline waters averaging around 10 °C at 3000 m in the Cape Basin and 20 °C at 1400 m in the Angola Basin. In comparison, mean bottom water temperatures in the Southern Ocean are lower, 5.5 °C and as low as 4 °C in the global mean. The deep Cape-Argentine and Angola basins also show the largest spread in simulated

temperatures and salinities across individual ensemble members (Fig. 4c-d). Zonal mean values differ by as much as 15 °C and a salinity of 5.8. Simulated uncertainties are highest in regions closest to the GBG and WR. Bottom water temperatures drop on average by 5-7 °C in response to the opening of these gateways (Fig. 5e-f), while a doubling of the atmospheric $CO_2$ increases temperatures by about 5 °C with a more homogeneous warming pattern (Fig. 5c). This makes $\Delta$WR and $\Delta$GBG the dominant control on temperature (Fig. 4e) and salinity (Fig. 4f) for the deep Angola Basin and some parts of the deep Cape-Argentine

Basin, respectively. Resulting bottom water temperatures at DSDP Site 361 (Fig. 4a) show a remarkably large range from 3.1 to 17.6 °C (Fig. 5a-b) with a mean value of 10.4 °C. The most constant temperatures with an ensemble spread below 4 °C are simulated for the upper 200 m just south of the WR. Southern Ocean temperatures are primarily controlled by $CO_2$, while enhanced advection of low-salinity waters resulting from a DP opening (see Supplementary Fig. S2) dominates the regional salinity variability (Fig. 4f).

The simulated temperature response to changing boundary conditions shows pronounced horizontal and vertical differences (Fig. 6). While $\Delta CO_2$ leads to a homogeneous warming of the upper Weddell Sea of 4-5 °C (Fig 6e), the northern Angola Basin shows an enhanced mean subsurface warming of 7.5 °C between 100-200 m (Fig. 6h). In contrast, subsurface warming just south of the WR (Fig. 6g) is reduced to below 2 °C in the mean with even a slight cooling at some grid points. The WR depth ($\Delta$WR) only affects the Angola Basin temperatures (Fig. 6h) and salinities. The spread in simulated salinities increases

considerably from south to north with values for the northern Angola Basin ranging between 36 and 42 (see Supplementary Fig. S3). An open DP reduces the temperature (Fig. 6e) and salinity in the Weddell Sea throughout the water column. The sea surface temperature (SST) drop of 4 °C for $\Delta$DP at DSDP Site 693 is of comparable magnitude as for $\Delta CO_2$. All ensemble members simulate a subsurface warming of 1-3 °C on the Falkland Plateau (Fig. 6f) and throughout the South Atlantic (Fig. 5g) in response to DP opening, caused by a higher production of warm and saline SAIW. This leads to simulated stage 2 bottom

water temperatures at DSDP Site 511 that are even slightly higher than those at the surface (Fig. 6b). The upper end of simulated bottom water temperatures on the Falkland Plateau coincides with paleotemperature estimates derived from oxygen isotope



**Figure 4.** Zonal mean sections of temperature and salinity for the South Atlantic and Southern Ocean. Values are averaged between 30°W and 10°W at the Falkland Plateau and between 30°W and 0° elsewhere. Panels show the (a-b) ensemble mean, (c-d) the difference between the minimum and maximum simulated value and (e-f) the respective process causing the largest mean change. Individual processes are defined in Section 2.3 and shown in Fig. 5 and Supplementary Fig. S2, respectively. Red lines show average water depth at the respective latitude and dotted lines in (e) indicate vertical sections analysed in Fig. 6.



**Figure 5.** Zonal mean temperature sections for the South Atlantic and Southern Ocean. Values are averaged between 30°W and 10°W at the Falkland Plateau and between 30°W and 0° elsewhere. Panels show the (a) minimum and (b) maximum simulated value across all 36 ensemble members and (c-e) the mean change associated with the individual processes defined in Section 2.3. Hatching indicates areas where all the respective 12 or 18 model responses agree on the sign of the change. Red lines show average water depth at the respective latitude.

data from benthic foraminifera (Huber et al., 2018) and belemnites (Price and Gröcke, 2002). Simulated SSTs range between 7-13 °C, while those derived from planktonic foraminifera (Huber et al., 2018) are higher by about 6 °C. Temperature estimates



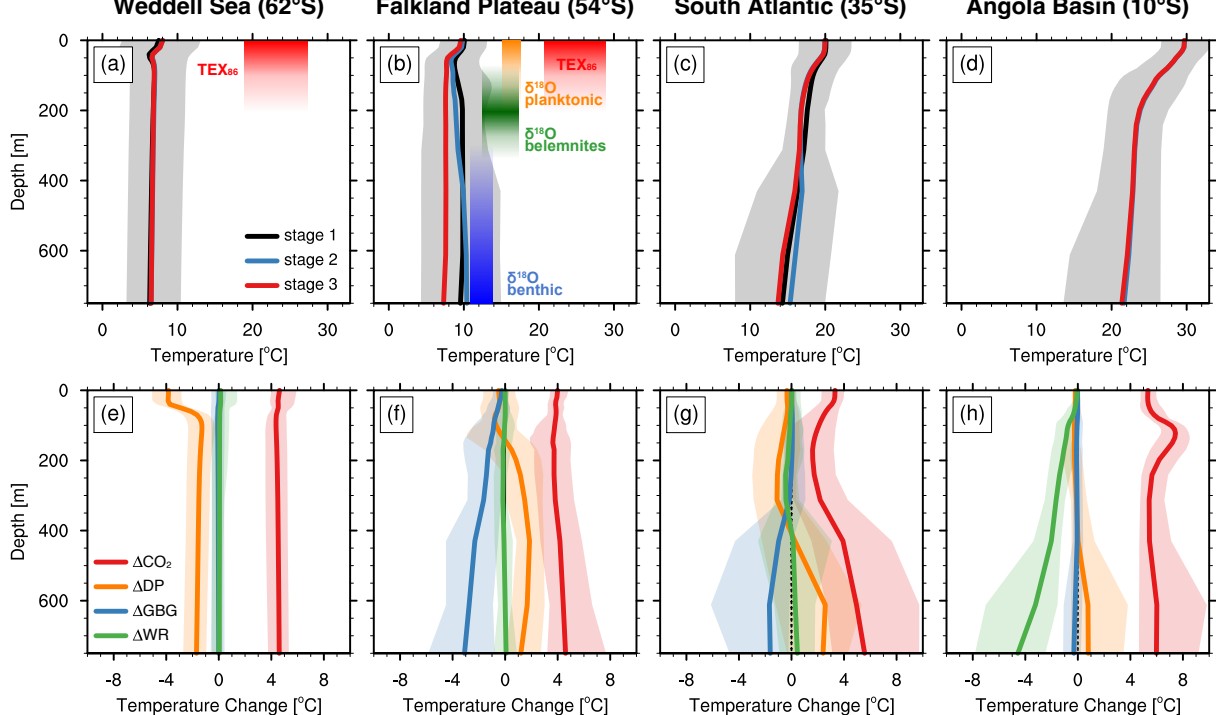

**Figure 6.** Simulated and reconstructed ocean temperatures across the Southern Ocean and South Atlantic. Solid lines represent regional averages (areas shown in Fig. 1f), while the shading indicates the simulated maximum and minimum grid point values of the respective area across all simulations. The upper row represents the temporal changes during stages 1 to 3, while the lower row shows the respective temperature changes caused by the individual processes defined in Section 2.3. The area used for averaging across the Falkland Plateau Basin in panels (b) and (f) moves westward for each consecutive model stage to allow comparison with the proxy record. Published ocean temperature estimates are based on TEX$_{86}$ (Jenkyns et al., 2012; Steinig et al., 2020) and oxygen isotope data from foraminifera (Huber et al., 2018) and belemnites (Price and Gröcke, 2002). Minimum TEX$_{86}$ temperatures are derived from a regional TEX$_{86}^H$ calibration following the discussion in (Steinig et al., 2020) and (Cavalheiro et al., 2021) for the Falkland Plateau and Weddell Sea, respectively, and are compared to maximum values from the global TEX$_{86}^H$ calibration (Kim et al., 2010). All available Aptian-Albian data are included and shown as $\pm 1$ standard deviation around the mean value at the approximate depth of signal formation. Foraminifera isotope data from the Aptian-Albian boundary interval at DSDP Site 511 are excluded due to a potential strong local freshwater influence (Huber et al., 2018).

based on the TEX$_{86}$ paleothermometer (Jenkyns et al., 2012; Steinig et al., 2020) exceed the ensemble mean temperatures in
the Weddell Sea and the Falkland Plateau by as much as 18 °C.

Vertical differences in the simulated responses of temperature and salinity influence the local water mass stratification (see Supplementary Fig. S3). An open DP significantly increases Weddell Sea stratification around 50 m water depth, while it is decreased on the Falkland Plateau. On average, $\Delta CO_2$ slightly enhances upper ocean stratification in the Weddell Sea and the South Atlantic. In the northern Angola Basin, on the other hand, higher $CO_2$ increases especially the surface salinity and
subsurface temperature, leading to a strongly reduced density stratification in the upper 100 m.





**Figure 7.** South Atlantic meridional overturning circulation (MOC). Panels show the (a) ensemble mean, (b) the respective boundary condition resulting in the largest mean MOC change, and (c-f) individual MOC changes associated with the processes defined in Section 2.3. Values are in Sv with 1 Sv = $10^6$ m$^3$/s. Positive (negative) values represent clockwise (counterclockwise) circulation. Hatching indicates regions where all ensemble members agree in terms of the sign of the circulation. Red lines show average water depth at the respective latitude. Dashed lines in (a) indicate vertical sections analysed in Fig. 8.



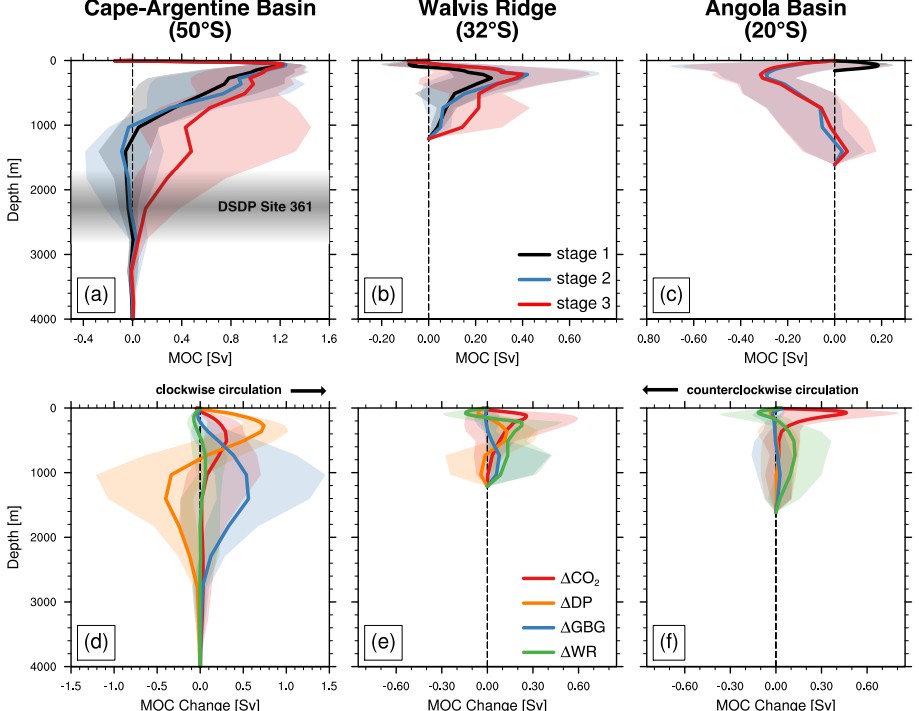

**Figure 8.** Profiles of the South Atlantic meridional overturning circulation (MOC). Profiles are shown for (left) 50°S, (middle) 32°S and (right) 20°S (see Fig. 7a). The upper row represents the temporal changes associated with stages 1 to 3, while the lower row shows the respective MOC changes caused by the individual processes defined in Section 2.3. Shading represents the respective ensemble spread with the ensemble mean indicated by solid lines. Values are in Sv with 1 Sv = $10^6$ m$^3$/s. Positive (negative) values represent clockwise (counterclockwise) circulation.

## 3.3 Meridional overturning circulation

We quantify basin-scale circulation differences in terms of changes in the meridional overturning circulation (MOC). Individual changes of the respective meridional and vertical velocity components are shown in Supplementary Figs. S4 and S5. The ensemble mean MOC in the Cape-Argentine Basin (Fig. 7a) is dominated by the salinity-driven intermediate water formation

described above. Positive values indicate northward flow at the surface, sinking at around 40°S and a subsequent southward return flow represented by SAIW. The mean volume transport of 1.2 Sv (1 Sv = $10^6$ m$^3$/s) is limited by the basin dimensions, and its maximum is reached within the upper 200 m. Despite the large range of applied boundary conditions, all 36 ensemble members agree on the sign of this overturning in the upper 500-1000 m of the southern South Atlantic (Fig. 8a), making this SAIW cell a persistent feature across all stages (see Supplementary Fig. S7). The vertical extent is limited to the Falkland

Plateau sill depth, leading to highly stagnant conditions in the deeper Cape-Argentine Basin.

  Further to the north, the WR cuts off the Angola Basin from the SAIW cell. Mean overturning in the Angola Basin is weak but driven by evaporative formation of dense waters just north of the WR (Fig. 3d) and upwelling at its northern end. The saline





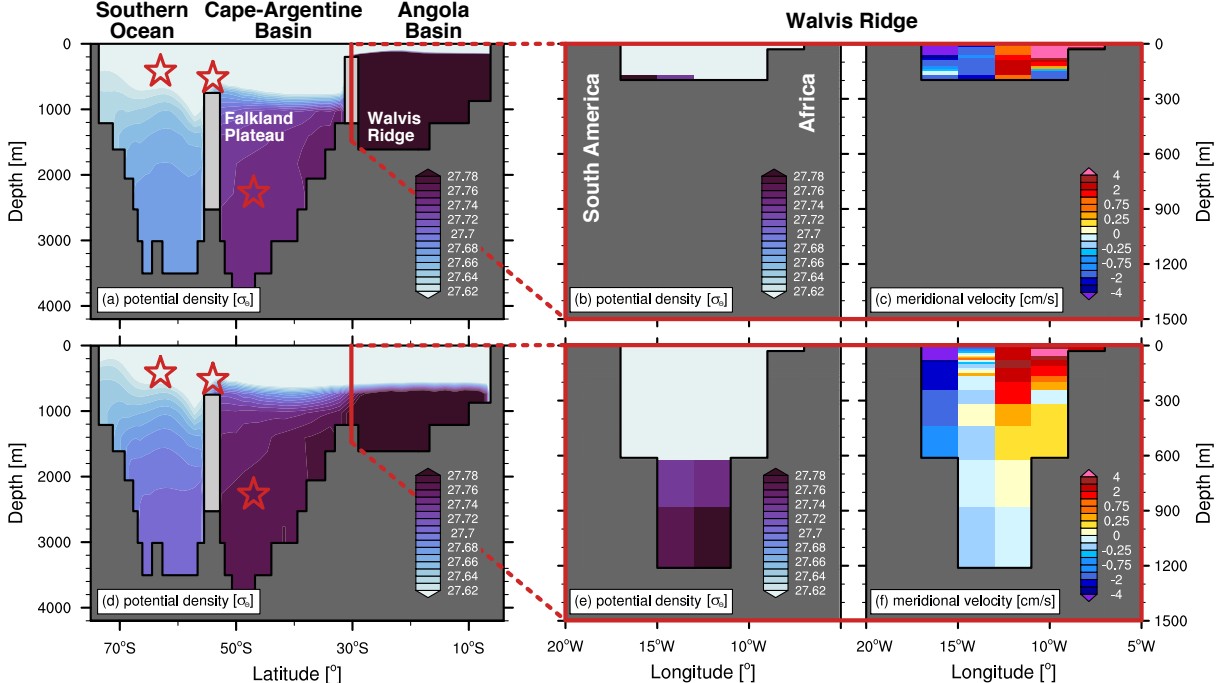

**Figure 9.** Export of Walvis Ridge Overflow Water (WROW). Panels show the stage 2 ensemble mean (a,d) zonal mean potential density as well as the (b,e) potential density and (c,f) meridional velocity across the Walvis Ridge. The upper row is averaged across the 6 ensemble members with a 200 m deep Walvis Ridge, the lower row shows results for the 6 ensemble members with a 1200 m water depth of the Walvis Ridge. Zonal mean values are averaged between 30°W and 10°W at the Falkland Plateau and between 30°W and 0° elsewhere. The color map for potential density only shows a small range of the data to focus on WROW density interval. The contour intervals for the meridional velocity are nonlinear.

and dense water filling the restricted Angola Basin eventually spills over the WR into the southern South Atlantic. This Walvis Ridge Overflow Water (WROW) is the densest water mass in the South Atlantic at the time and reaches the deepest parts of the
Cape-Argentine Basin (Fig. 9), even for a shallow WR. Depth-integrated volume transport across the WR is weak and does not exceed 0.7 Sv (Fig. 8b), but it is always directed southward, i.e., export into the southern South Atlantic, for the densest water in the bottom layer (Fig. 9c,f). The circulation in the upper 200 m of the Angola Basin is highly sensitive to $\Delta CO_2$ (Fig. 8f). The northern end of the Angola Basin during stages 2 and 3 reaches the tropical regions (Fig. 1d,e) leading to locally enhanced precipitation rates and river runoff (Fig. 3c). This net surface freshwater input decreases the surface density and limits vertical
mixing in the northern part of the basin at 600 ppmv $CO_2$. The doubling of $CO_2$ increases evaporation rates and, surprisingly, also reduces local precipitation (see Supplementary Fig. S6). Evaporation in the northernmost Angola Basin strongly exceeds precipitation at 1200 ppmv $CO_2$, while precipitation dominates at 600 ppmv $CO_2$ north of 10°S (Fig. 10c,d). The resulting increased surface salinities at higher $CO_2$ lead to denser surface waters and enable a shallow and clockwise MOC (Fig. 7c)





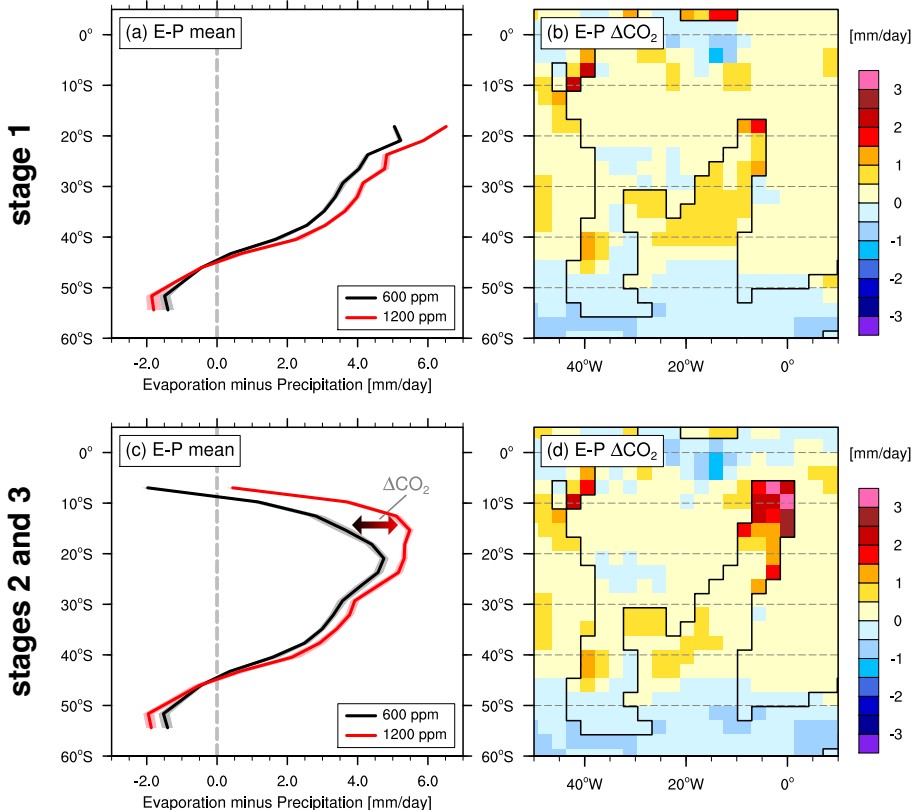

**Figure 10.** South Atlantic freshwater balance. (a, c) Annual mean fluxes of evaporation minus precipitation zonally averaged over the South Atlantic and grouped by atmospheric $CO_2$ concentration for the two different basin geometries. Solid lines represent the ensemble mean of all (a) n=6 and (c) n=12 experiments with the ensemble spread indicated by the shading around the mean. (b, d) Maps of annual mean $\Delta CO_2$, i.e. the average change of the freshwater balance due to a doubling of atmospheric $CO_2$. Positive values indicate a net freshwater flux from the ocean to the atmosphere. Global fields of precipitation and evaporation are shown in Supplementary Fig. S6.

associated with enhanced subsurface temperatures between 100 and 200 m depth (Fig. 6h) and a reduced density stratification. 245 In contrast, the smaller Angola Basin during stage 1 is rather insensitive to freshwater balance changes (Fig. 10b).

As the regional overturning is driven by strong evaporation, the doubling of $CO_2$ leads to an overall amplification and downward extension of the SAIW cell (Fig. 7c). The opening of the DP creates the largest change of the upper ocean MOC (Fig. 7b) with a mean increase of 0.7 Sv between 200-400 m (Fig. 7d). The largest change of the MOC in the deeper Cape-Argentine Basin is simulated for the opening of the GBG (Fig. 7b) as the positive SAIW cell during stage 3 extends down to 250 the new maximum sill depth of 2500 m (Fig. 7e). The response to $\Delta GBG$ is positive for all ensemble members (Fig. 8d), but the ensemble spread is large. The strength of the stage 3 deep cell in the South Atlantic is significantly reduced for an open DP (Fig. 8d). Overall, total southward volume transport of 0.3 Sv (range from 0.1-0.8 Sv) for depths below 2000 m is very low.



WROW export increases for stage 3 (Fig. 8b) as this dense water mass contributes to the enhanced South Atlantic deep water exchange.

## 4 Discussion

Our ensemble approach reveals a fundamental control of the applied boundary conditions on the simulated regional circulation, while their relative contributions vary considerably between individual basins. Furthermore, we show that the simulated response to a forcing factor, e.g. $\Delta CO_2$, can be highly sensitive to the respective mean state and study region. In the following, we will discuss the dominant controls on the simulated circulation in two key regions and assess potential implications for the interpretation of local proxy records. In the end, we compare the analysed physical mechanisms with the proposed gateway opening and carbon burial history presented in Dummann et al. (2020).

### 4.1 Angola Basin sensitivity to atmospheric $CO_2$

Strong evaporation and resulting elevated surface densities drive the local overturning circulation in the restricted Angola Basin. Highest surface densities and intermediate water formation are simulated just north of the Walvis Ridge between 20-25°S because local precipitation gradually increases towards the Intertropical Convergence Zone (ITCZ). This is consistent throughout all simulations and in agreement with simulated Early Albian mixed layer depths in Uenzelmann-Neben et al. (2017). Ensemble members with more restricted basin geometries indicate that the entire basin, except for a thin surface layer, was filled with this dense water mass with maximum salinities exceeding 42. This agrees with biomarker reconstructions indicating a hypersaline environment (S > 40) for large parts of the water column during the Aptian (Naafs and Pancost, 2014; Behrooz et al., 2018). While the applied WR depth influences absolute temperatures and salinities in the deeper parts of the Angola Basin, we find a dominant control of atmospheric $CO_2$ on the circulation and related water column stratification for the upper 200 m during stages 2 and 3. These two later stages represent an Early Albian Angola Basin geometry with a northern extent up to 6°S (Sewall et al., 2007). Local precipitation exceeds evaporation in this part of the basin which leads to locally reduced surface salinities and densities. Enhanced river runoff from tropical Africa amplifies the net surface freshwater gain, leading to an estuarine circulation with a southward surface flow in the low $CO_2$ simulations. A somewhat surprising result is that the doubling of atmospheric $CO_2$ does not only increase evaporation over the ocean but also reduces local precipitation, contradicting a simple dry gets drier, wet gets wetter paradigm. In our simulations, the northern Angola Basin is located at the southern edge of the ITCZ. An increase of $CO_2$ leads to a narrowing of the ITCZ and therefore a reduction in local precipitation over the Angola Basin. The drier conditions further increase the positive evaporation-precipitation balance over the entire Angola Basin, but particularly in the northern part. The higher surface densities drive a shallow, anti-estuarine circulation in the upper 100-200 m. Resulting increased mixing of heat and fresh water into the subsurface in turn diminishes the vertical density gradient in the northern Angola Basin.

Due to the restricted circulation, the magnitude of these stratification changes also depend on the strength of local river runoff and resulting surface freshening. While the ensemble mean salinity in the southern Angola Basin exceeds 40, it is



reduced to 38-39 at its northern boundary. Chaboureau et al. (2012) show that this north-south gradient can increase sharply for a narrower Late Aptian basin geometry, elevated orography or changed freshwater routing, with minimum surface salinities of as low as 20. This implies that our simulated influence river runoff on the northernmost basin is a conservative estimate. The proposed mechanism and its influence on upper ocean stratification may therefore have been even more important during earlier, narrower stages of basin evolution. In support of that, Pérez-Díaz and Eagles (2017) showed that, within the reported

uncertainty ranges, seafloor spreading in the Angola Basin could have reached latitudes of about 10°S as early as 120 Ma.

The sensitivity of local water column stratification to atmospheric changes has the potential to influence the reported Aptian black shale deposition (Stein et al., 1986; Bralower et al., 1994; Naafs and Pancost, 2014). Importantly, the proposed mechanism leads to an increased stratification in the Angola Basin at lower $CO_2$ concentrations, contrary to the results simulated for the other basins. Behrooz et al. (2018) showed that organic carbon burial in the Angola Basin exhibits a pronounced cyclicity

and conclude that orbitally driven variations in the hydrological cycle caused changes in nutrient availability and upper ocean stratification that ultimately controlled local black shale deposition. In addition, we show that similar changes in environmental conditions can also be caused by variations in atmospheric $CO_2$. Wallmann et al. (2019) demonstrated that oscillations of atmospheric $CO_2$ at similar amplitude as used in our study could arise entirely internally, driven by a marine redox seesaw. This would provide a mechanism for changes in water column stratification in the Angola Basin in addition to orbitally driven

changes.

Our results add to the idea of a dominant influence of atmospheric dynamics on Early Cretaceous marine black shale deposition (Wagner et al., 2013). We show that the simulated turnover in upper ocean circulation and stratification can be explained by large-scale shifts of the atmospheric circulation and a northward migration of tropical precipitation driven by atmospheric $CO_2$ changes. We note that due to the prescribed land surface, our simulations potentially underestimate the full

response of the hydrological cycle due to missing vegetation-atmosphere feedbacks of greenhouse climates (e.g., Upchurch et al., 1998; Levis et al., 2000; Zhou et al., 2012). Uncertainties in local orography and related changes in precipitation (Houze, 2012) are additional factors with demonstrated effects on the equatorial Aptian climate (Chaboureau et al., 2012). Nevertheless, the simulated narrowing of the ITCZ is a large-scale pattern across the Aptian-Albian South American and African continents. Furthermore, projections for the twenty-first century also show a consistent narrowing and intensification of the ITCZ in

response to elevated atmospheric $CO_2$ levels (Byrne et al., 2018), which is supported by the observational record (Wodzicki and Rapp, 2016). We conclude that the progressive northward opening of the Early Cretaceous Angola Basin not only changed the mean freshwater balance but also enhanced the regional sensitivity to global climate variability.

### 4.2  Comparison to temperature reconstructions

Independent validation of the simulated temperature distribution is difficult due to scarce quantitative proxy estimates for the

study region and interval. To our knowledge, the only suitable Aptian-Albian oxygen isotope or $TEX_{86}$ data come from the Weddell Sea and Falkland Plateau (see overview in O'Brien et al., 2017). For the latter, absolute $\delta^{18}O$-derived surface temperatures from Aptian-Albian planktonic foraminifera (Huber et al., 2018) average 16 °C, which is 3.0 °C higher than maximum model SSTs. As the Falkland Plateau SSTs are mainly sensitive to $\Delta CO_2$, an unrealistic further doubling of atmospheric $CO_2$





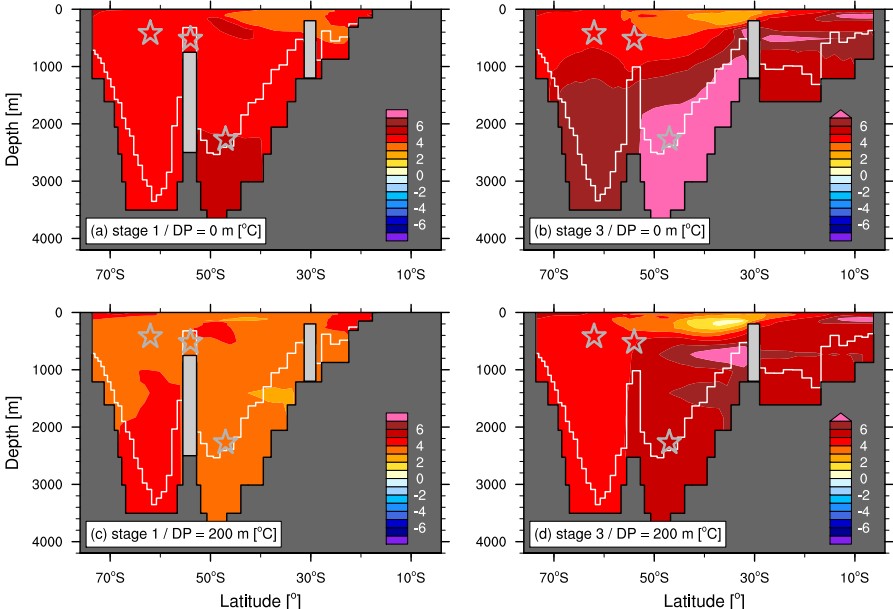

**Figure 11.** Zonal mean sections of temperature changes associated with a $CO_2$ doubling averaged across the South Atlantic and Southern Ocean. Panels show $\Delta CO_2$ for stages 1 and 3 for (a,b) a closed DP and (c,d) for a shallow DP opening,. White lines show average water depth at the respective latitude and stage.

concentrations to $\sim 2400$ ppmv would be needed to close the model-proxy misfit. Another possible solution would be an
enhanced hydrological cycle with an associated lower $\delta^{18}O$ of local seawater that would reduce reconstructed temperatures (Huber et al., 2018). In accordance to this, our 1200 ppmv $CO_2$ simulations show that the net fresh water gain (precipitation exceeding evaporation) between 50-60°S was on average 24% larger for the Cretaceous than in a comparable present-day simulation with the same model. The excess freshwater is advected across the Falkland Plateau as part of the surface limb of the described South Atlantic overturning circulation and may have influenced local carbonate chemistry and temperature re-
construction. We note that exceptionally low $\delta^{18}O$ at the Falkland Plateau have also been reported for the Turonian greenhouse climate (Bice et al., 2003).

As uncertainties in the local freshwater balance would primarily affect absolute temperature reconstructions for surface-dwelling species, we infer that the model-proxy comparison for subsurface and bottom water temperatures are a more robust test. The warmest bottom water temperatures on the Falkland Plateau are simulated for high $CO_2$, an open DP and a closed
GBG. In addition to the direct radiative warming in response to higher $CO_2$ concentrations, we show that all three factors lead to an enhanced subsurface advection of warm and saline SAIW into the Falkland Basin. This creates a subsurface temperature maximum during stage 2, when bottom water temperatures exceed SSTs by up to 3 °C. The upper end of simulated bottom water temperatures of up to 15 °C agrees with oxygen isotope-derived temperatures from Aptian-Albian benthic foraminifera (Huber et al., 2018) and Aptian belemnites (Price and Gröcke, 2002). Importantly, a closed DP reduces subsurface temperatures



on the Falkland Plateau by up to 3 °C due to a reduction in the strength of SAIW formation. The resulting average temperatures, especially at lower $CO_2$ levels, are therefore not consistent with the independent bottom water temperature reconstructions and our simulations indicate an at least shallow water connection across the DP as a more likely scenario.

Similiar to the Anogala Basin, the $\Delta CO_2$ temperature changes in the southern South Atlantic also vary over time, with overall higher sensitivity and larger spatial variability during stages 2 and 3 (Fig. 11). Furthermore, our ensemble simulations 340 reveal that the $\Delta CO_2$ temperature response is also modulated by the DP configuration. While changes in the hydrological cycle control the Angola Basin $\Delta CO_2$ response, the width of the DP influences the local ocean dynamics in the southern South Atlantic via changes in the SAIW overturning strength. Both mechanisms illustrate the highly nonlinear interaction between paleogeographic and radiative forcing mechanisms, which can only be fully captured by an ensemble approach. Our analysis shows that the export of SAIW during the early opening phase was confined to the Falkland Basin and the advection of this 345 warm and salty water mass led to an inversion of the local vertical temperature gradient. Furthermore, changes in the Early Cretaceous paleogeography alone likely caused substantial temperature variations on the Falkland Plateau (Lunt et al., 2016). The special oceanographic setting of this region therefore implies that conclusions from local temperature records should only be translated to global climatic changes with caution.

The difference between model-predicted and $TEX_{86}$-reconstructed (Jenkyns et al., 2012; Steinig et al., 2020) temperatures 350 of up to 18 °C can not be reconciled, even considering the large calibration uncertainties (O'Brien et al., 2017). The specific $TEX_{86}$ export dynamics in the restricted Early Cretaceous Atlantic Ocean may have influenced the local $TEX_{86}$-temperature relation (Steinig et al., 2020). Applying a regional $TEX_{86}^{H}$ calibration from the modern Mediterranean and Red Sea (Kim et al., 2015; Steinig et al., 2020) reduces absolute reconstructed temperatures on the Falkland Plateau by $\sim$6 °, thereby reducing, but not eliminating, the model-data misfit. In accordance with Steinig et al. (2020), who used the same model but with a different 355 paleobathymetry, our analysis shows remarkable similarities between the Aptian-Albian South Atlantic and the present-day Mediterranean Sea. Both are deep and restricted evaporative basins characterised by a salinity driven, anti-estuarine circulation. Even the observed mean temperature of $\sim$13 °C and salinities above 38 (Borghini et al., 2014) are within the simulated Cretaceous ensemble range. Since the $TEX_{86}$ warm bias is also found in the Mediterranean Outflow Water along the Portuguese continental margin (Kim et al., 2016), this region could be a potential modern analogue to partly explain the high $TEX_{86}$-360 derived temperatures for the Early Cretaceous Falkland Plateau. The same mechanism has been used to explain model-data discrepancies and to improve consistency with other proxy techniques in the Valanginian Weddell Sea (Cavalheiro et al., 2021).

### 4.3 Integrated ocean circulation and carbon burial history

Our ensemble results allow to test and to complement the hypotheses on the opening history of the Early Cretaceous South Atlantic presented in Dummann et al. (2020). Their reconstruction of regional water mass circulation and mixing is primarily 365 based on the isotopic differences in seawater $\epsilon_{Nd}(t)$ between more radiogenic SAIW ($\epsilon_{Nd}(t)$ signatures between -3.2 to -2.5) and less radiogenic Pacific-sourced waters ($\epsilon_{Nd}(t)$ signatures between -5.5 to -3.5). Therefore, one of our most important results is that the SAIW formation mechanism and region are consistent throughout all 36 ensemble members, i.e. radiogenic SAIW formation and export are independent of the boundary condition uncertainties used in this study. This result is also in agreement



with the simulated formation of Atlantic Intermediate Water in a different climate model (Uenzelmann-Neben et al., 2017). As even the deeper WR significantly limits lateral water mass exchange with the Angola Basin, SAIW is formed in the area of local maxima in E-P fluxes in the northwestern Cape-Argentine Basin. The proposed radiogenic isotopic signature of SAIW, potentially derived from erosional input of the Paraná large igneous province (Fig. 3a), is therefore a plausible end member for tracing the regional water mass mixing during the Aptian to Albian. Due to the restricted and narrow basin, the overall mean volume transport of 1.2 Sv is an order of magnitude lower than the strength of the present-day Atlantic Meridional Overturning Circulation (McCarthy et al., 2015). In the following, we use the model-derived temporal changes in regional ocean circulation to complement the new stratigraphic framework and local gateway history (Fig. 12).

The relatively unradiogenic $\epsilon_{Nd}$(t) values in the Weddell Sea (DSDP Site 249) and the Falkland Basin (DSDP Sites 327 and 511) during the latest Barremian to Early Aptian (∼123 to ∼117 Ma) are consistent with the simulated eastward advection of South Pacific water masses via the DP (Fig. 12c). During this time, the Falkland Basin formed a restricted, semi-enclosed basin that was shielded by the African continent in the north, favouring organic-rich black shale deposition (Dummann et al., 2021a). Due to the limited grid resolution and uncertain local paleobathymetry, the model is not able to resolve the proposed restricted local circulation on the Falkland Plateau itself. However, the overall stronger influence of Southern Ocean Water for a more eastward position of the Falkland Plateau is clearly demonstrated. Black shales with total organic carbon (TOC) contents of up to 20% are also found at DSDP Site 361 in the deep Cape Basin (Dummann et al., 2021b). In the model, the deep South Atlantic circulation is sluggish, but influenced by saline Walvis Ridge Overflow Water (WROW) from the north. The combination of the most restricted environment with the highest evaporation and lowest precipitation forms the largest reservoir of dense water masses in the Angola Basin. In the model, this water mass eventually overflows the WR and sinks along the topography into the deepest parts of the Cape-Argentine Basin. Average WROW volume transport of ∼0.1 Sv is low and vigorous entrainment of South Atlantic waters reduces density differences on its way down the slope. The model results agree with earlier hypotheses that anoxic waters from the Angola Basin spilling over the WR increased stratification in the deep South Atlantic and promoted marine organic carbon burial (e.g., Natland, 1978; Arthur and Natland, 1979; Stein et al., 1986). Unradiogenic $\epsilon_{Nd}$(t) signatures at site 361 support a disconnection of the deep basin from the overlying SAIW cell and could indicate an influence of the unusually unradiogenic Nd signatures of -12 found north of the WR (Dummann et al., 2023). We conclude that the production of WROW is physically plausible and consistent with available geochemical data, although it is still under debate how much, if any, water mass exchange was actually possible across the Walvis Ridge barrier (e.g. Heine et al., 2013; Pérez-Díaz and Eagles, 2017; Cui et al., 2023; Dummann et al., 2023).

TOC values at site 361 decrease during stage 2 (Fig. 12b), indicating an enhanced ventilation of the deep Cape Basin between ∼117 to ∼113 Ma. Our idealised simulations reveal no significant changes in the region between stages 1 and 2, but the applied paleogeographic changes are by design minimal. We therefore provide two hypotheses that could explain the reconstructed shift to a better ventilated, more radiogenic water mass in the deep Cape Basin. First, our simulations show that the depth of the SAIW cell was ultimately controlled by the sill depth of the Falkland Plateau. During the Aptian-Albian, the Falkland Plateau subsided from an initial shelf to an outer shelf to middle bathyal environment (see Section 2.2). Our simulations suggest that the maximum SAIW depth would have deepened in parallel, thereby increasing the oxygen content and radiogenic $\epsilon_{Nd}$(t) signature



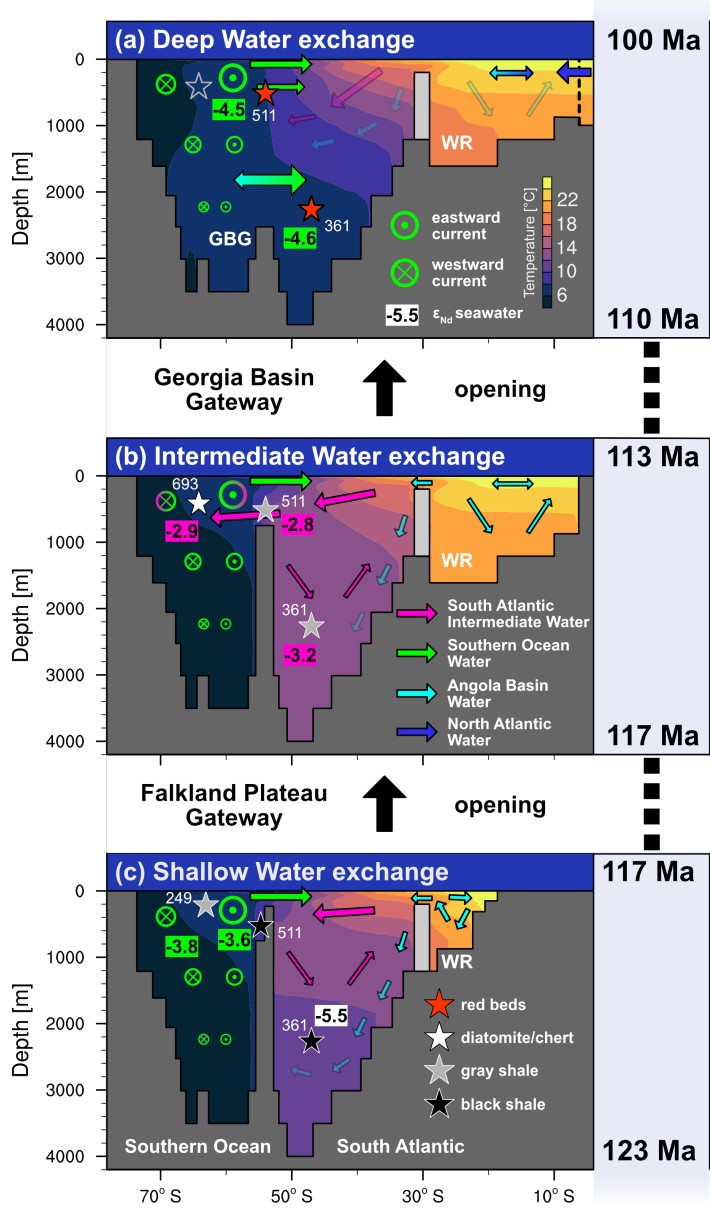

**Figure 12.** Schematic of the proposed integration of the Aptian to Albian circulation evolution of the South Atlantic and Southern Ocean. Contours represent ensemble mean zonal mean (30°W-0°) ocean temperature for (a) stage 3, (b) stage 2 and (c) stage 1. Arrows only qualitatively depict the proposed circulation. The colour code of the star symbols represents the main lithology at individual study sites, while the bold numbers show the average $\epsilon_{Nd}(t)$ isotope signature for the respective time periods as reported in Dummann et al. (2020) and shown in Fig. 1. The northern boundary of the Angola Basin at ~5°S in (a) has been modified to visualise proposed onset of intermediate water exchange via Equatorial Atlantic gateway (Dummann et al., 2023).




of the deeper South Atlantic. Second, due to the large uncertainty in paleobathymetric estimates of the Early Cetaceous South

Atlantic (Pérez-Díaz and Eagles, 2017), we used a constant Cape-Argentine Basin geometry for all three stages. In reality, the progressive opening would have led to a more gradual increase of the basin volume and, potentially, in the strength of the SAIW overturning cell. Furthermore, assuming that the advection of WROW was limited by the WR sill depth and relatively constant from the Early to the Late Aptian, the relative influence of locally formed and better ventilated water masses would have increased compared to the poorly-ventilated WROW. In either case, these Late Aptian paleogeographic changes enabled

a less restricted circulation in the deep Cape Basin. At the same time, intermediate water sites on the Falkland Plateau and in the Weddell Sea indicate the advection of more radiogenic SAIW because of the opening of the Falkland Plateau Gateway (Dummann et al., 2020, 2021a).

The opening of the Georgia Basin Gateway between ∼113 and ∼110 Ma represents the most profound shift in both the geochemical data and the ensemble simulations. Our model results show that the outflowing SAIW was confined to the south-

western tip of Africa for all stages. As the Falkland Plateau moves further to the west during stage 3, subsurface circulation in the entire Falkland Basin switched from an export of SAIW to an influx of cold Southern Ocean water, which is consistent with a gradual shift in $\epsilon_{Nd}$(t) in the Falkland Basin, reflecting an inflow of less radiogenic, Pacific-sourced water masses (Fig. 12a). The simultaneous onset of red bed deposition across the South Atlantic provides complementary evidence for an enhanced advection of cold, well-oxygenated water masses sourced at high-latitudes . Data for DSDP Site 361 indicate the

presence of a similarly unradiogenic and oxygenated southern-sourced water mass in the deep Cape Basin during the Albian. In contrast, the simulations are characterised by a weak southward export of South Atlantic waters via the whole GBG depth during stage 3. We identify two processes that probably acted in concert as the most probable explanation for this difference. First, our simulated stage 3 is only represented by a minimal opening of the GBG with a width of only two model grid points. An even less restricted deep-water exchange at a later time would likely lead to an increase in the contribution of northward

flowing Southern Ocean deep waters. In accordance with this hypothesis, Donnadieu et al. (2016) show that the deep South Atlantic during the Cenomanian/Turonian boundary at ∼94 Ma was already characterised by a net northward transport of 4.2 Sv between 1,200 and 2,750m. Second, we show that the enhanced salinity-driven formation of intermediate and deep waters in the South Atlantic was a direct consequence of the highly restricted environment of the young basins. In case of any water mass exchange across the WR, the gradual opening and deepening of the equatorial gateway to the North Atlantic would have

increased horizontal water mass exchange, reduced surface salinities, and therefore suppressed intermediate and deep water formation in the South Atlantic (Poulsen et al., 2001). Dummann et al. (2023) show that intermediate water mass exchange across the Equatorial Atlantic gateway started at 107 Ma and even deep water exchange was already possible by ∼100 Ma. This could have led to a reduction or termination of SAIW and WROW production and a gradual transition from a previously salinity-driven deep-water formation at low latitudes to a temperature-driven convection regime in the high-latitude Southern

Ocean (Poulsen et al., 2001; Donnadieu et al., 2016).





## 5    Summary and Conclusions

We present an ensemble approach to deal with large uncertainties in deep time paleo boundary conditions. Instead of dedicated sensitivity experiments, we use an ensemble of 36 climate model experiments to simulate the Barremian to Albian opening of the South Atlantic Ocean under full consideration of the uncertainties in available boundary conditions. This study serves as a test case to demonstrate the feasibility and benefits of such an approach to assess the fundamental controls on the Early Cretaceous ocean circulation and their temporal changes in the evolving South Atlantic. Through close integration of a regional stratigraphic framework and available geochemical data we conclude that:

1. Excessive evaporation and restricted circulation led to the salinity-driven formation of intermediate waters in the early evolutionary stages of the subtropical South Atlantic under all boundary conditions. The southward export of these warm and saline intermediate waters was balanced by the wind-driven advection of cold and fresh high-latitude Southern Ocean waters and can be used to track water mass exchange with the Southern Ocean.

2. Increasing geographical restrictions towards the northern end of the South Atlantic produced densest water masses in the silled Angola Basin. These saline and presumably anoxic waters may have entered the South Atlantic via overflow across the Walvis Ridge and potentially promoted the deposition of organic-rich black shales in the deep Cape Basin during the Early Aptian.

3. Paleogeography was at least as important as $CO_2$ in driving temperature and salinity changes for most parts of the restricted South Atlantic and must therefore be considered in any reconstruction of the local temperature evolution. Limitations in available boundary conditions lead to Aptian-Albian ocean temperature uncertainties of up to 15 °C, especially in the deep Cape-Argentine and Angola basins.

4. The consecutive opening of marine gateways in the South and equatorial Atlantic led to a stepwise transition from the initial influence of warm, oxygen-depleted intermediate and deep waters produced at subtropical latitudes, to a dominance of high-latitude convection forming cold and oxygenated deep waters in the Southern Ocean. This turnover in regional deep water ventilation ultimately controlled the end of enhanced organic carbon burial during the Early Albian.

5. Our ensemble approach reveals significant nonlinearities between individual forcing mechanisms. Progressive northward rifting enhanced the sensitivity of the Angola Basin circulation and stratification to changes in atmospheric $CO_2$ via large-scale shifts in tropical precipitation. The DP modulates the mean strength of the meridional SAIW cell and its sensitivity to $\Delta CO_2$, with an improved model data fit for subsurface temperatures on the Falkland Plateau for a shallow to intermediate DP depth. These interrelated changes in atmospheric and ocean dynamics cannot be identified with traditional single sensitivity experiments and highlight the benefits of an ensemble approach.



*Data availability.* All model data used to generate the figures in this publication will be made available through the Zenodo open repository (https://zenodo.org, last access: 11 November 2023).

**Information about the Supplement**

Supplementary Figures S1 to S7 are available in the file "supplement-Controls_on_Early_Cretaceous_South_Atlantic-Steinig_et_al.pdf".

*Author contributions.* PH, SF, and TW conceived the overall project and acquired funding. SS, SF and WP designed the modelling experiments. SS carried out the simulations and led the analysis. WD, PH, MF and TW led the interpretation of the geochemical data. SS wrote the manuscript with contributions from all authors.

*Competing interests.* The authors declare that they have no conflict of interest.

*Acknowledgements.* We thank the German Research Foundation (DFG) for funding this research within the project "Evolving carbon sinks in the young South Atlantic: Drivers of global climate in the early Cretaceous greenhouse?" (grant numbers FL378/1-1, FL378/1-2 and HO2188/9). S.F. has been additionally supported by the German Research Foundation through collaborative research project SFB 754 (sub-project A7). Model integrations were conducted at the Computing Center of Kiel University. We thank Jens Herrle for helping with the interpretation of the geochemical data and Stefan Hagemann for constructing the parameters for the hydrological discharge model.



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
