# Peer review of "Controls on Early Cretaceous South Atlantic Ocean circulation and carbon burial - a climate model-proxy synthesis"

_EGUsphere, 2023_

## Author Comment (AC1)

*Controls on Early Cretaceous South Atlantic Ocean circulation and carbon burial -*
*a climate model-proxy synthesis*

**Response to Reviewer #1**

We thank reviewer #1 for their overall very positive feedback and for their time to provide very constructive comments to improve the quality of the overall manuscript. Below we provide a point-by-point reply to all comments raised. Reviewer comments are in black, our response is in red font.

In their manuscript "Controls on Early Cretaceous South Atlantic Ocean circulation and carbon burial – a climate model-proxy synthesis", Steinig and co-authors present a large ensemble of Early Cretaceous (Barremian-Albian) simulations specifically designed to investigate how some key tectonic features control the ocean circulation, and ultimately, carbon burial in the restricted and expanding South Atlantic basin, in which evidence for Early Cretaceous black shales, documenting intense carbon burial and seafloor anoxia, has been found. Such evidence has notably been discussed by the same group of authors in recent papers (e.g. Dummann et al. 2021) and this earth system model investigation is a logical follow-up.

They notably find that the paleogeographic changes throughout the Early Cretaceous led to a shift from restricted basins in the South Atlantic filled by locally-formed warm and saline intermediate and deep waters to increased exchange with the Southern Ocean, thereby filling the Cape-Argentine and Angola Basins with fresher, colder and more oxygen-rich waters. This scenario is consistent with a progressive shift from intense carbon burial in these restricted basins in the Early Aptian to more oxygenated conditions in the Aptian, in agreement with geochemical data.

The authors also show that their evaluation of paleogeographic uncertainties with an ensemble approach, which considers joint changes in boundary conditions, highlights significant non-linearities in the sensitivity of the South Atlantic ocean circulation to said changes. For instance, the sensitivity of the Angola Basin to $CO_2$ increases with its progressive northward expansion. This ensemble approach also allows the authors to demonstrate that small tectonic changes can lead to very large changes in local water mass properties, which need to be taken into account in the interpretation of the geological record.

I enjoyed reading this manuscript. The methodology is sound and adequately presented. The results are robustly demonstrated and interesting and the figure are clear and informative. I only have minor comments, which should be straightforwardly answered. Nice job!

We highly appreciate the positive feedback and the detailed and accurate summary of our work. Thank you!

Comments
Orbital parameters. Though an impressive set of simulations has been performed, I think that stating that a "full" exploration of the uncertainties has been carried out is a bit exaggerated. Recently, Sarr et al. (2022) demonstrated significant orbital variability of the ocean circulation and anoxia. I think the manuscript should be revised to include some discussion about this.

We thank the reviewer for pointing us to this highly relevant paper and we agree that these results should be discussed in our revised manuscript. The Sarr et al. (2022) results are a very valuable addition to discuss how our simulated oceanographic mean state and changes might have been related to changes of the carbon burial in the developing South Atlantic. We will therefore add their results to the discussion of the black shale deposition in the Angola Basin (Section 4.1, l. 294) and to the overall framework of the carbon burial history in the South Atlantic (Section 4.3). Beyond this

relevance for the local water mass oxygenation and carbon burial, we would still argue that our approach covers the main boundary condition uncertainties that dominate temperature, salinity and circulation in the early Cretaceous South Atlantic. Sarr et al. (2022) point out that orbitally-driven circulation changes are largest in deep water formation regions in the southern Pacific and mainly impact the South Atlantic by changing the oxygenation state of the advected intermediate and deep waters. Reported changes in sea surface salinities in the Atlantic sector are also generally smaller than signals discussed in our study (< 0.4 psu). In the revised manuscript, we will therefore better highlight that our ensemble approach is designed to capture variability/uncertainty in regional temperature, salinity and circulation, while the background oxygenation and its sensitivity to orbital variability shown by Sarr et al. (2022) are important to deduce and discuss any resulting changes in regional carbon burial.

Figure 1. Consider adding "Angola Basin" and "Cape Basin" for clarity.

We will include these basin labels in Fig. 1 as suggested.

l. 103-107. It is unclear what portion of the land/sea mask and bathymetry is updated using Sewall et al. (2007). Is it the full Southern Ocean and South Atlantic, with the latter being subsequently refined according the specified sensitivity tests? Or is it only the expanding Southern Atlantic? Please reformulate.

This formulation is indeed not very clear. The updated sentence will read:
"we replaced the regional land-sea mask and ocean bathymetry of the study area, i.e. the whole region shown in Fig. 1f, with published climate model boundary conditions for the Early Cretaceous (Sewall et al., 2007)."

Figure 2f. It is hard to agree that the deep ocean in the DP closed simulations is close to equilibrium because it looks like it is gaining buoyancy. There are chances that further integration will lead to destratification of the deep ocean and overturning. Also it might be useful to show in Supplementary the spin-up of the global deep ocean because closing DP may impact the Pacific as well and we do not know whether the global deep ocean has reached equilibrium.

Yes, integration length for individual ensemble members is limited due to the computational effort of the ensemble approach. Arguably, the changes in the Drake Passage depth have the largest influence on the local and global deep ocean conditions. We therefore extended the integrations for three ensemble members with three different Drake Passage depths by a further 1500-2000 model years each (1500 years for the closed Drake Passage, 2000 model years each for the 200 m and 1400 m depth scenarios) to check for the long-term stability of our results. The updated spin-up figure is attached as Supplementary Fig. 1 to this document and will replace Fig. 2b,d,f in the revised manuscript. The results show that the change in ocean temperature during the spin-up extension of these three ensemble members is negligible over the study region, even for the deep ocean. We therefore argue that the overall spin-up length of our ensemble members is justified and does not influence our results.

We agree that showing the global spin-up of our sensitivity experiments is a very useful suggestion (which was also made by reviewer #2) and add this plot (Supplementary Fig. 2 of this document) to the Supplementary Material of the revised manuscript. The figure shows that the existing ensemble members are already near equilibrium – even in the deep ocean – as the temperature drift over the additional 1500-2000 model years of all extended simulations is below 0.3 °C at all levels.

Section 3. It is unfortunate that no large-scale description of the ocean circulation is given before entering details about the South Atlantic. This is not given either in Steinig et al. (2020). At a minimum,

you should provide details about the deep-water formation zones and global overturning. Do any of these change with altering the SA or the Southern Ocean (Drake) bathymetry? Consider also discussing larger ocean circulation changes with respect to orbital variability as shown by Sarr et al. (2022), see above.

We agree that a description of the large-scale ocean circulation and its sensitivity to boundary condition changes would be very helpful to put the regional circulation changes into context. This has also been requested by reviewer #2. In our simulations, changes in the Falkland Plateau and Georgia Basin gateways, as well as the Walvis Ridge do not influence global ocean dynamics significantly, due to the limited export of local water masses. But we do find an important influence of the Drake Passage depth on the large-scale ocean circulation and the associated meridional heat transport. We use the three extended ensemble members described above to show the influence of the Drake Passage depth on the global ocean circulation. These results are shown in the attached Supplementary Fig. 3. As the Drake Passage depth decreases (panels b and c), the reduced volume transport is compensated by an equally large increase in the advection of warm and saline subtropical waters between Africa and India. Overall, surface temperatures increase in the Southern Hemisphere and decrease in the Northern Hemisphere, with maxima of +/- 3 °C in the polar regions. In the revised manuscript, we will include Supplementary Fig. 3 of this document into the main manuscript and start the Results part of the manuscript with a new section (3.1 Global ocean circulation), which briefly describes the main features of the global ocean circulation (meridional overturning circulation and areas of deep water formation) and their sensitivity to the Drake Passage depth. We will also compare these findings to the orbitally-driven changes in deep water formation described in Sarr et al. (2022).

l. 186. The subsurface maximum is visible for salinity but much less clear for temperature (Fig. 4a).

We will remove the reference to a subsurface temperature maximum.

l. 198-199. The GBG and CO2-driven anomalies in salinity are of the same magnitude or even larger than that of DP on Fig. S2.

This is true for the South Atlantic, but l. 198-199 were meant to focus on the Southern Ocean only. We will change the wording to make this clearer:
*"Southern Ocean temperatures are primarily controlled by CO2, while enhanced advection of low-salinity waters resulting from a DP opening (see Supplementary Fig. S2) dominates the salinity variability **in the Southern Ocean** (Fig. 4f). "*

l. 208-209. The subsurface extension affected by warming in response to Drake Passage opening is not the same for the Falkland Plateau (200-800 m) and the South Atlantic (~ 400 – 800 m, not throughout as stated). Also it is Fig. 6g, not 5g.

We agree and will change this to:
*"All ensemble members simulate a subsurface warming of 1-3 °C on the Falkland Plateau (**200-800 m;** Fig. 6f) and in the South Atlantic (**400-800 m;** Fig. **6**g) in response to DP opening"*

Figure 7. How variable is the simulated MOC within a single simulation? Judging by its amplitude, and the fact that agreement across all ensemble members occurs over limited regions, I would say quite highly.

We are not sure whether we understand this comment correctly. Does the question refer to the year-to-year variability of the overturning strength? If so, we illustrated the interannual variability for a single simulation in Supplementary Fig. 4. The strength of the main clockwise (positive) circulation cell is substantially higher than its interannual standard deviation. The mean strength of the cell is ~1.5 Sv

with a standard deviation of 0.3 Sv and an overall amplitude of change of +/- 40% around the mean value. This shows that the overturning cell is a consistent feature of the South Atlantic circulation.

Figure 7 / SAIW. Is the intermediate water formation in the Cape Basin somehow driven by downwelling in the center of the subtropical gyre defined in Fig. 3? Is the convection there seasonal?

Yes, the intermediate water formation is confined to the center of the gyre, indicated by the local mixed layer maximum between 30-40°S in Fig. 3e. The convection shows a strong seasonality with a peak during the months July to September and a complete suppression during austral summer (new Supplementary Fig. 5 in this document). We will add the information about the location and seasonality to the revised manuscript to make this clearer to the reader.

l. 243. Fig. 7c (and d,e,f) shows difference in MOC intensity but it is unclear whether you computed the difference between absolute MOC amplitude or not? Could you find a way to show in which regions the MOC changes sign? Another example is on Fig. 7d. $\Delta$MOC is -0.3 Sv at 50°S and 1500 m depth but the absolute mean MOC at this location is 0.15 Sv on Fig. 7a. Does it mean the MOC changes sign there? I don't think so but it is rather unclear.

We agree that the calculation needs to be explained more clearly. We indeed calculate the differences between the absolute MOC fields, e.g. the average of all 1200 ppm simulations minus the average of all 600 ppm simulations. We will clarify this in the revised manuscript. We also agree with the reviewer that it is difficult to directly relate this to changes in the sign of the overturning. To make this information available to the expert reader, we will add another Figure to the Supplementary Material of the revised manuscript. This will contain the same sections as currently in Fig. 7c-f, but will show the percentage of ensemble members that change sign due to each of the four sensitivity parameters (CO2, DP, GBG, WR). For this, we will compare the sign of the MOC directly between each pair of the ensemble where only the relevant parameter differs, instead of comparing ensemble means. For example, this results in 18 pairs of model simulations that only differ in the amount of CO2. We will test for each pair and grid point, whether the sign of the MOC is different between both ensemble members and sum up the number of sign changes for each grid point. This will identify areas in the South Atlantic that change their direction of the prevailing overturning circulation and the related mechanism.

Figure 10. Are the contours are the E-P computed over the ocean only?

Yes, the zonal mean E-P fluxes in Fig. 10a,c are averaged over the ocean only. We will update the figure caption to highlight this choice.

l. 277. "contradicting a simple dry gets drier, wet gets wetter paradigm". That the $\Delta CO_2$ E-P be different between stage 1 and stage 2-3 over continental areas is hardly visible on Fig. 10. And the mean $\Delta(P+E)$ over subtropical South America and Africa on Fig. S6h between 1200 and 600 ppm looks quite similar to me as well.

Indeed, the 'dry gets drier, wet gets wetter' (DDWW) paradigm is mainly based on oceanic data and cannot easily be applied to land areas (e.g. Greve et al., 2014). For our current study, we are focusing on the marine regions and our own simulations follow the DDWW for most of the global ocean surface (Fig. S6h). The edges of the ITCZ - and therefore the most northern grid points of the Angola Basin - are different and change from a previously precipitation to now evaporation-dominated region in response to a CO2 doubling. We will clarify that we are discussing the marine realm only in the revised manuscript.

l. 297-298. Not sure that the reference to Wallmann et al. (2019) is relevant. Their mechanism has been demonstrated for the mid-Cretaceous, with a significantly deeper configuration of the Central American seaway and more restricted North Atlantic than in the Early Cretaceous paleogeography used here. It is unclear whether it would apply in the Early Cretaceous.

The difference between the early and mid-Cretaceous paleogeography of the Atlantic Ocean is indeed an important limitation for a direct comparison. Nevertheless, we argue that at least discussing the possibility of internally generating pCO2 variability in the restricted Cretaceous ocean basins (in addition to the well-known orbital variability) is justified. In the revised manuscript we will add a sentence to also highlight the limitations of this comparison.

l. 345. Is there any evidence for a downward positive temperature gradient in the proxy record?

To our knowledge, there is no convincing evidence. Huber et al. (2018) point out an anomaly in their Falkland Plateau (DSDP Site 511) temperature reconstructions during the "Aptian/Albian Boundary Interval", where local bottom water temperature reconstructions exceed surface temperatures (their Fig. 7). They acknowledge that they can't find a satisfying explanation for this, but focusing just on this single data anomaly would be speculative. We rather argue that many surface temperature proxies seem to incorporate a significant contribution of subsurface signals into their sedimentary evidence. An unusual subsurface warming as in our simulations might therefore be one possibility to partly explain the extremely warm surface temperature reconstructions at such a high latitude of ~54°S.

l. 420-421. Unclear. Fig.12a (stage 3) shows (northward) export of southern-sourced waters across the GBG at the depth of DSDP Site 361 but it is stated that "the simulations are characterized by a weak southward export of SA waters via the whole GBG depth during stage 3".

Yes, this is indeed a discrepancy between the geochemical and simulation data. We do acknowledge this and provide two hypotheses to explain this difference in the rest of the paragraph (l.422-435). In addition, the sentence "the simulations are characterized by a weak southward export of SA waters via the whole GBG depth during stage 3" refers to the net export across the whole gateway, which is calculated as the volume difference between southward and northward flow at different grid points and depth levels. The simulated bottom water flow does show a northward transport of southern-sourced waters into the South Atlantic. This is also visible due to the presence of the cold bottom water temperatures around Site 361 during stage 3 (Fig. 12a), compared to the warm South Atlantic-sourced waters during stage 2 (Fig. 12b). We will update the description in l. 420-421 and the arrow in Fig. 12a in the revised manuscript.

l. 427-434. As it stands, change deep-water formation by intermediate-water formation. Fig. 3e shows max MLD of ~ 200 m, which suggests that convection occurs across a limited depth. It is very possible that dense waters are formed in shallow coastal areas and that the density anomaly is then propagated on the seafloor by the bottom boundary layer scheme of NEMO. But I am not sure the latter is convection as people generally think. Alternatively, it would be interesting to output daily variables to see if the MLD monthly max of ~ 200 m is the result of one or a couple of deeper convective episodes intertwined with the absence of convection. In this case, the term "deep-water formation" would probably be more justified.

This is a very valid point. We agree that we don't simulate open-ocean convection down to the seafloor in the South Atlantic and admit that this needs to be highlighted. We argue that there were potentially two relevant mechanisms. Convection (< 200 m) of warm and saline waters in the subtropical gyre of the South Atlantic and formation of very saline and dense waters around the shelf regions and particularly in the Angola Basin. These were the densest locally-formed water masses and might have entered the Cape-Argentine Basin as overflow waters, following down the topography

of the Walvis Ridge (Fig. 9). We think it is important to discuss these two mechanisms here but will clarify in the text that the bottom waters were not formed by convective cells (in contrast to the high-latitude Pacific Ocean), but rather by sluggish and dense overflow waters from the northernmost shelf regions.

References:

Greve, P., Orlowsky, B., Mueller, B. *et al.* Global assessment of trends in wetting and drying over land. *Nature Geosci* **7**, 716–721 (2014). https://doi.org/10.1038/ngeo2247

Huber, B. T., MacLeod, K. G., Watkins, D. K., and Coffin, M. F.: The rise and fall of the Cretaceous Hot Greenhouse climate, Global and
Planetary Change, 167, 1–23 (2018), https://doi.org/10.1016/j.gloplacha.2018.04.004

[Figure]

**Supplementary Figure 1:** Time series of annual mean ocean temperatures in °C averaged over the South Atlantic and Southern Ocean (0°-90°S, 40°W-20°E) at 5 m, 732 m and 2290 m. Color coding represents the different stages defined in the main text and the two levels of atmospheric CO2.

[Figure]

**Supplementary Figure 2:** Time series of globally averaged annual mean ocean temperatures in °C at 5 m, 732 m and 2290 m water depth.

[Figure]

**Supplementary Figure 3:** Ocean circulation and surface temperature sensitivity to proto-Drake Passage depth. Vectors show ocean velocity in cm/s averaged over the upper 100 m on contours of 2 m temperature in °C. Numbers indicate the depth integrated volume transport across the respective sections. Values are in Sv with 1 Sv = $10^6$ m³/s. Panels show (a) annual mean values for the original DP depth of 1400 m and the respective differences to this for the (b) 200 m and (c) closed DP. Velocity vector length is limited to two times the reference magnitude for figure clarity. Results are shown for three individual ensemble members with an extended spin-up to reach a global equilibrium state. Values are averaged over model years 4900-4999.

[Figure]

**Supplementary Figure 4:** South Atlantic meridional overturning circulation variability. Results are shown for a 100-year window of a single ensemble member (stage 3, 1400 m Drake Passage, 1200m Walvis Ridge, 1200ppm CO2) for illustration purpose. Panels show the (a) time mean, (b) interannual standard deviation and (c) time series of the maximum overturning of the domain. All values are in Sv.

[Figure]

**Supplementary Figure 5:** Annual cycle of South Atlantic mixed layer depth. Results are shown for a single ensemble member (stage 3, 1400 m Drake Passage, 1200m Walvis Ridge, 1200ppm CO2) for illustration purpose. Values are in m.

---

## Author Comment (AC2)

*Controls on Early Cretaceous South Atlantic Ocean circulation and carbon burial -*
*a climate model-proxy synthesis*

**Response to Reviewer #2**

We thank reviewer #2 for their positive evaluation of our work and for providing very helpful suggestions for further improvement. Below we provide a point-by-point reply to all comments raised. Reviewer comments are in black, our response is in red font.

In their manuscript *Controls on Early Cretaceous South Atlantic Ocean circulation and carbon burial – a climate model-proxy synthesis'* by S. Steinig et al. used an ensemble of 36 coupled ocean-atmosphere simulations to evaluate South Atlantic Ocean circulation during the early Cretaceous. They setup the simulations in order to account for the uncertainty in basin configuration, regional seaways (Drake passage and Walvis ridge) and pCO2. To do so they account for all possible combination of forcing. They then carefully analyzed the contribution of each factor on oceanographic changes in the Southern Atlantic basin.  In the last part of the paper, they integrate simulated scenarios with eNd data and sedimentological information to discuss evolution of Southern Ocean oceanography during the Barremian-Aptian.They notably highlight that paleogeography changes, happening at higher temporal resolution than those commonly investigated with paleo-climate modeling might have very strong impact locally, which would be reflected in the proxy.

I find the methodology very interesting. Those ensemble simulations, that are now accessible with classic ESM/GCM thanks to improved model performance seems to be one of the steps forward for deep time paleo-climate modeling, though they are not often used yet. The choice of boundary conditions and forcing are well justified in the text and relevant with regard to existing literature. The manuscript is overall super easy to read, and I appreciate the care that have been given to figures. I really enjoyed reading this paper and find remarkable that I am almost left with nothing to say. So congrats!

Thank you for the thorough summary and the positive feedback regarding the study design.

I only have minor comments:

- Authors analyze the dynamics of Southern Atlantic as isolated from the global circulation. I wonder whether the boundary conditions they implemented have also an impact of global circulation and whether global changes could affect the regional signal. For example, do the water masses transported through Drake Passage and entering the Southern Atlantic always have the same characteristics? Or does changes in source water characteristics might also impact the paleoceanography of the Southern Atlantic basin?

This has also been recommended by reviewer #1 and we agree with the usefulness of a description of the mean state and sensitivity of the global ocean circulation. In our simulations, we do find an important influence of the Drake Passage depth on the large-scale ocean circulation and the associated meridional heat transport. For this, we extended the simulations of three ensemble members with three different Drake Passage depths for a further 1500-2000 model years each to analyse changes in the global equilibrium circulation. Please see the new Supplementary Fig. 3 and description of the main results in our response to reviewer #1.

Figure 1 – 2 : It could be useful to have the location of basins indicated on the maps as well so it's easier to visualize. While it's not essential for understanding what the authors are writing it would help reader not always familiar with local geology structures to better understand.

We will add basin labels to Figure 1 to have a common reference for all basin and gateway names used throughout the study.

Figure 1 : Color distinction between sites 249 and 693 is not strong enough.

Agreed. The colour contrast between both sites will be increased.

Figure 2 : would be good to have a plot showing global time series for the sensitivity as well.

Yes, we will add this plot (Supplementary Fig. 1 in this document) to the Supplementary Material of the revised manuscript and present the results in Section 2.3. This plot now also includes the three extended spin-up integrations to test for deep ocean equilibrium mentioned above. It shows that the existing ensemble members are already near equilibrium – even in the deep ocean – as the temperature drift over the additional 1500-2000 model years of all extended simulations is below 0.3 °C at all levels.

Figure 4c-d : This one is a very minor suggestion but changing the color scale for c) and d) to a scale that have various shade of the same color (like white – light blue to dark blue for example) would help better emphasize where temperature/salinity are highly variable and where variability is small.

We agree that this would help to highlight areas of small/high variability and will update the colour scales accordingly.

L.185 – Note that the maxima in salinity and temperature are not strictly located at the same depth: one looks to be in the subsurface while the other is at the surface in the figure. Also *"the southern South Atlantic and the Angola Basin both show pronounced"* – do the authors mean Cape-Argentine basin instead of southern Atlantic ? Because this would make more sense in the sentence?

Yes, we will change this to "Cape-Argentine Basin" and remove the reference to a subsurface temperature maximum. The part will now read:
*"As a result of the salinity-driven intermediate water formation, the Cape-Argentine and the Angola basins both show a pronounced subsurface maximum in salinity at depths between 200-600 m (Fig. 4b)*."

L. 197-198 – authors should also refer to figure 5c in this sentence.

This figure reference will be added as suggested.

L.275 – *"somewhat surprising result is that the doubling of atmospheric CO2 does not only increase evaporation over the ocean but also reduces local precipitation, contradicting a simple dry gets drier, wet gets wetter paradigm."* This sounds that it is a general response while looking at figure S6 at first order region of E-P > (<) 0 have increased (decreased) E-P with CO2 doubling no matter if this is driven more by changes in precipitation or evaporation. I suggest that the authors reformulate this sentence.

Yes, the sentence indeed intended to highlight that the northern Angola Basin behaves differently than the zonal mean, as it switches from a precipitation to an evaporation-dominated region between the 600 ppm and 1200 ppm simulations. We will split this sentence into two to make this contrast clearer. The part will now read:
"*A somewhat surprising result is that the doubling of atmospheric CO2 does not only increase evaporation over the ocean but also reduces local precipitation over the most northern grid points of the Angola Basin, resulting in a switch from a previously precipitation to now evaporation-dominated*

*region in response to a CO2 doubling. This is different compared to the dry gets drier, wet gets wetter pattern simulated for most of the ocean surface (Supplementary Fig. S6)."*

L337 - *Anagola Basin* --> Angola Basin ?

Yes, this will be changed.

L385 - Walvis Ridge Overflow Water – can you plot that on you maps/cross section to improve readability?

The label and an arrow will be added to the sections shown in Fig. 9.

Figure S1 : to a horizontal grid of 0.5◦ x 0.5◦ horizontal grid  --> to a horizontal grid of 0.5◦ x 0.5◦ . Please also specified that this is the case for Perez-Diaz & Eagles, 2017 but that KCM plots are at model resolution.

The sentence will be changed to:
*"Only the minimum and maximum reported water depths are shown for each stage. KCM bathymetries are shown on the native model grid, while the reconstructions have been re-gridded to a 0.5° x 0.5° horizontal grid.*

Figure S6 vs Figure 10. Please chose between P-E or E+P notation.

We will remove this inconsistency by changing Supplementary Fig. S6 to show E-P fluxes as well.

[Figure]

**Supplementary Figure 1:** Time series of globally averaged annual mean ocean temperatures in °C at 5 m, 732 m and 2290 m water depth.

---

## Author Response (AR1)

*Controls on Early Cretaceous South Atlantic Ocean circulation and carbon burial -*
*a climate model-proxy synthesis*

**Response to Reviewer #1**

We thank reviewer #1 for their overall very positive feedback and for their time to provide very constructive comments to improve the quality of the overall manuscript. Below we provide a point-by-point reply to all comments raised. Reviewer comments are in black, our response is in red font. Line and figure numbers in our responses are based on the revised manuscript.

In their manuscript "Controls on Early Cretaceous South Atlantic Ocean circulation and carbon burial – a climate model-proxy synthesis", Steinig and co-authors present a large ensemble of Early Cretaceous (Barremian-Albian) simulations specifically designed to investigate how some key tectonic features control the ocean circulation, and ultimately, carbon burial in the restricted and expanding South Atlantic basin, in which evidence for Early Cretaceous black shales, documenting intense carbon burial and seafloor anoxia, has been found. Such evidence has notably been discussed by the same group of authors in recent papers (e.g. Dummann et al. 2021) and this earth system model investigation is a logical follow-up.

They notably find that the paleogeographic changes throughout the Early Cretaceous led to a shift from restricted basins in the South Atlantic filled by locally-formed warm and saline intermediate and deep waters to increased exchange with the Southern Ocean, thereby filling the Cape-Argentine and Angola Basins with fresher, colder and more oxygen-rich waters. This scenario is consistent with a progressive shift from intense carbon burial in these restricted basins in the Early Aptian to more oxygenated conditions in the Aptian, in agreement with geochemical data.

The authors also show that their evaluation of paleogeographic uncertainties with an ensemble approach, which considers joint changes in boundary conditions, highlights significant non-linearities in the sensitivity of the South Atlantic ocean circulation to said changes. For instance, the sensitivity of the Angola Basin to $CO_2$ increases with its progressive northward expansion. This ensemble approach also allows the authors to demonstrate that small tectonic changes can lead to very large changes in local water mass properties, which need to be taken into account in the interpretation of the geological record.

I enjoyed reading this manuscript. The methodology is sound and adequately presented. The results are robustly demonstrated and interesting and the figure are clear and informative. I only have minor comments, which should be straightforwardly answered. Nice job!

We highly appreciate the positive feedback and the detailed and accurate summary of our work. Thank you!

Comments
Orbital parameters. Though an impressive set of simulations has been performed, I think that stating that a "full" exploration of the uncertainties has been carried out is a bit exaggerated. Recently, Sarr et al. (2022) demonstrated significant orbital variability of the ocean circulation and anoxia. I think the manuscript should be revised to include some discussion about this.

We thank the reviewer for pointing us to this highly relevant paper and we agree that these results should be discussed in our revised manuscript. The Sarr et al. (2022) results are a very valuable addition to discuss how our simulated oceanographic mean state and changes might have been related to changes of the carbon burial in the developing South Atlantic. We therefore added their results to the discussion of the black shale deposition in the Angola Basin (l. 325-328) and to the

overall framework of the carbon burial history in the South Atlantic (l. 409-414). Beyond this relevance for the local water mass oxygenation and carbon burial, we would still argue that our approach covers the main boundary condition uncertainties that dominate temperature, salinity and circulation in the early Cretaceous South Atlantic. Sarr et al. (2022) point out that orbitally-driven circulation changes are largest in deep water formation regions in the southern Pacific and mainly impact the South Atlantic by changing the oxygenation state of the advected intermediate and deep waters. Reported changes in sea surface salinities in the Atlantic sector are also generally smaller than signals discussed in our study (< 0.4 psu). In the revised manuscript, we therefore better highlighted that our ensemble approach is designed to capture variability/uncertainty in regional temperature, salinity and circulation, while the background oxygenation and its sensitivity to orbital variability shown by Sarr et al. (2022) are important to deduce and discuss any resulting changes in regional carbon burial.

Figure 1. Consider adding "Angola Basin" and "Cape Basin" for clarity.

These basin labels have been added to Fig. 1 as suggested.

l. 103-107. It is unclear what portion of the land/sea mask and bathymetry is updated using Sewall et al. (2007). Is it the full Southern Ocean and South Atlantic, with the latter being subsequently refined according the specified sensitivity tests? Or is it only the expanding Southern Atlantic? Please reformulate.

This formulation is indeed not very clear. The updated sentence (l. 103-105) now reads:
"we replaced the regional land-sea mask and ocean bathymetry of the study area, i.e. the whole region shown in Fig. 1f, with published climate model boundary conditions for the Early Cretaceous (Sewall et al., 2007)."

Figure 2f. It is hard to agree that the deep ocean in the DP closed simulations is close to equilibrium because it looks like it is gaining buoyancy. There are chances that further integration will lead to destratification of the deep ocean and overturning. Also it might be useful to show in Supplementary the spin-up of the global deep ocean because closing DP may impact the Pacific as well and we do not know whether the global deep ocean has reached equilibrium.

Yes, integration length for individual ensemble members is limited due to the computational effort of the ensemble approach. Arguably, the changes in the Drake Passage depth have the largest influence on the local and global deep ocean conditions. We therefore extended the integrations for three ensemble members with three different Drake Passage depths by a further 1500-2000 model years each (1500 years for the closed Drake Passage, 2000 model years each for the 200 m and 1400 m depth scenarios) to check for the long-term stability of our results. The spin-up Figure 2 has been updated accordingly in the revised manuscript. The results show that the change in ocean temperature during the spin-up extension of these three ensemble members is negligible over the study region, even for the deep ocean. We therefore argue that the overall spin-up length of our ensemble members is justified and does not influence our results.

We agree that showing the global spin-up of our sensitivity experiments is a very useful suggestion (which was also made by reviewer #2) and add this plot to the Supplementary Material of the revised manuscript (new Figure S2). The figure shows that the existing ensemble members are already near equilibrium – even in the deep ocean – as the temperature drift over the additional 1500-2000 model years of all extended simulations is below 0.3 °C at all levels.

These results have been added to the Methods section (l. 166-173) of the revised manuscript.

Section 3. It is unfortunate that no large-scale description of the ocean circulation is given before entering details about the South Atlantic. This is not given either in Steinig et al. (2020). At a minimum, you should provide details about the deep-water formation zones and global overturning. Do any of these change with altering the SA or the Southern Ocean (Drake) bathymetry? Consider also discussing larger ocean circulation changes with respect to orbital variability as shown by Sarr et al. (2022), see above.

We agree that a description of the large-scale ocean circulation and its sensitivity to boundary condition changes would be very helpful to put the regional circulation changes into context. This has also been requested by reviewer #2. In our simulations, changes in the Falkland Plateau and Georgia Basin gateways, as well as the Walvis Ridge do not influence global ocean dynamics significantly, due to the limited export of local water masses. But we do find an important influence of the Drake Passage depth on the large-scale ocean circulation and the associated meridional heat transport. We use the three extended ensemble members described above to show the influence of the Drake Passage depth on the global ocean circulation. These results have been added as the new Figure 3 of the main manuscript. As the Drake Passage depth decreases (panels b and c), the reduced volume transport is compensated by an equally large increase in the advection of warm and saline subtropical waters between Africa and India. Overall, surface temperatures increase in the Southern Hemisphere and decrease in the Northern Hemisphere, with maxima of +/- 3 °C in the polar regions. In the revised manuscript, we added a new section at the start the Results part of the manuscript (3.1 Global ocean circulation), which briefly describes the main features of the global ocean circulation (areas of deep water formation and meridional overturning circulation) and their sensitivity to the Drake Passage depth (l. 176-190).

l. 186. The subsurface maximum is visible for salinity but much less clear for temperature (Fig. 4a).

We have removed the reference to a subsurface temperature maximum (l. 208-209).

l. 198-199. The GBG and $CO_2$-driven anomalies in salinity are of the same magnitude or even larger than that of DP on Fig. S2.

This is true for the South Atlantic, but this sentence was meant to focus on the Southern Ocean only. We have changed the wording to make this clearer (l. 220-222):
*"Southern Ocean temperatures are primarily controlled by CO2, while enhanced advection of low-salinity waters resulting from a DP opening (see Supplementary Fig. S2) dominates the salinity variability **in the Southern Ocean** (Fig. 4f). "*

l. 208-209. The subsurface extension affected by warming in response to Drake Passage opening is not the same for the Falkland Plateau (200-800 m) and the South Atlantic (~ 400 – 800 m, not throughout as stated). Also it is Fig. 6g, not 5g.

We agree and have added this clarification (l. 230-232):
*"All ensemble members simulate a subsurface warming of 1-3 °C on the Falkland Plateau (**200-800 m;** Fig. 6f) and in the South Atlantic (**400-800 m;** Fig. **6**g) in response to DP opening"*

Figure 7. How variable is the simulated MOC within a single simulation? Judging by its amplitude, and the fact that agreement across all ensemble members occurs over limited regions, I would say quite highly.

We are not sure whether we understand this comment correctly. Does the question refer to the year-to-year variability of the overturning strength? If so, we illustrated the interannual variability for a single simulation in Supplementary Fig. S1 of this document. The strength of the main clockwise

(positive) circulation cell is substantially higher than its interannual standard deviation. The mean strength of the cell is ~1.5 Sv with a standard deviation of 0.3 Sv and an overall amplitude of change of +/- 40% around the mean value. This shows that the overturning cell is a consistent feature of the South Atlantic circulation.

Figure 7 / SAIW. Is the intermediate water formation in the Cape Basin somehow driven by downwelling in the center of the subtropical gyre defined in Fig. 3? Is the convection there seasonal?

Yes, the intermediate water formation is confined to the center of the gyre, indicated by the local mixed layer maximum between 30-40°S in Fig. 3e. The convection shows a strong seasonality with a peak during the months July to September and a complete suppression during austral summer (new Supplementary Fig. S2 in this document). We added this information about the location and seasonality to the revised manuscript (l. 200-202).

l. 243. Fig. 7c (and d,e,f) shows difference in MOC intensity but it is unclear whether you computed the difference between absolute MOC amplitude or not? Could you find a way to show in which regions the MOC changes sign? Another example is on Fig. 7d. $\Delta$MOC is -0.3 Sv at 50°S and 1500 m depth but the absolute mean MOC at this location is 0.15 Sv on Fig. 7a. Does it mean the MOC changes sign there? I don't think so but it is rather unclear.

We agree that the calculation needs to be explained more clearly. We indeed calculate the differences between the absolute MOC fields, e.g. the average of all 1200 ppm simulations minus the average of all 600 ppm simulations. We clarified this in the revised manuscript (l. 268-272). We also agree with the reviewer that it is difficult to directly relate this to changes in the sign of the overturning. To make this information available to the expert reader, we added the new Figure S9 to the Supplementary Material of the revised manuscript. This contains the same sections as currently in Fig. 7c-f, but shows the percentage of ensemble members that change sign due to each of the four sensitivity parameters (CO2, DP, GBG, WR). For this, we compare the sign of the MOC directly between each pair of the ensemble where only the relevant parameter differs, instead of comparing ensemble means. For example, this results in 18 pairs of model simulations that only differ in the amount of CO2. We test for each pair and grid point, whether the sign of the MOC is different between both ensemble members and sum up the number of sign changes for each grid point. This identifies areas in the South Atlantic that change their direction of the prevailing overturning circulation and the related mechanism.

Figure 10. Are the contours are the E-P computed over the ocean only?

Yes, the zonal mean E-P fluxes in Fig. 11a,c are averaged over the ocean only. We have updated the figure caption accordingly.

l. 277. "contradicting a simple dry gets drier, wet gets wetter paradigm". That the $\Delta CO_2$ E-P be different between stage 1 and stage 2-3 over continental areas is hardly visible on Fig. 10. And the mean $\Delta(P+E)$ over subtropical South America and Africa on Fig. S6h between 1200 and 600 ppm looks quite similar to me as well.

Indeed, the 'dry gets drier, wet gets wetter' (DDWW) paradigm is mainly based on oceanic data and cannot easily be applied to land areas (e.g. Greve et al., 2014). For our current study, we are focusing on the marine regions and our own simulations follow the DDWW for most of the global ocean surface (Supplementary Fig. S7h). The edges of the ITCZ - and therefore the most northern grid points of the Angola Basin - are different and change from a previously precipitation to now evaporation-dominated region in response to a CO2 doubling. We have clarified that we are discussing the marine realm only in the revised manuscript (l. 302-306).

l. 297-298. Not sure that the reference to Wallmann et al. (2019) is relevant. Their mechanism has been demonstrated for the mid-Cretaceous, with a significantly deeper configuration of the Central American seaway and more restricted North Atlantic than in the Early Cretaceous paleogeography used here. It is unclear whether it would apply in the Early Cretaceous.

The difference between the early and mid-Cretaceous paleogeography of the Atlantic Ocean is indeed an important limitation for a direct comparison. Nevertheless, we argue that at least discussing the possibility of internally generating pCO2 variability in the restricted Cretaceous ocean basins (in addition to the well-known orbital variability) is justified. In the revised manuscript we added a note about the limitation of this comparison (l. 330-331).

l. 345. Is there any evidence for a downward positive temperature gradient in the proxy record?

To our knowledge, there is no convincing evidence. Huber et al. (2018) point out an anomaly in their Falkland Plateau (DSDP Site 511) temperature reconstructions during the "Aptian/Albian Boundary Interval", where local bottom water temperature reconstructions exceed surface temperatures (their Fig. 7). They acknowledge that they can't find a satisfying explanation for this, but focusing just on this single data anomaly would be speculative. We rather argue that many surface temperature proxies seem to incorporate a significant contribution of subsurface signals into their sedimentary evidence. An unusual subsurface warming as in our simulations might therefore be one possibility to partly explain the extremely warm surface temperature reconstructions at such a high latitude of ~54 °S.

l. 420-421. Unclear. Fig.12a (stage 3) shows (northward) export of southern-sourced waters across the GBG at the depth of DSDP Site 361 but it is stated that "the simulations are characterized by a weak southward export of SA waters via the whole GBG depth during stage 3".

Yes, this is indeed a discrepancy between the geochemical and simulation data. We do acknowledge this and provide two hypotheses to explain this difference in the rest of the paragraph (l.460-475). In addition, the sentence "the simulations are characterized by a weak southward export of SA waters via the whole GBG depth during stage 3" refers to the net export across the whole gateway, which is calculated as the volume difference between southward and northward flow at different grid points and depth levels. The simulated bottom water flow does show a northward transport of southern-sourced waters into the South Atlantic. This is also visible due to the presence of the cold bottom water temperatures around Site 361 during stage 3 (Fig. 13a), compared to the warm South Atlantic-sourced waters during stage 2 (Fig. 13b). We updated the description of this discrepancy (l. 458-460) and the arrow in Fig. 13a in the revised manuscript to indicate that we see our simulation results as only the very first step towards the advection of southern-sourced water masses.

l. 427-434. As it stands, change deep-water formation by intermediate-water formation. Fig. 3e shows max MLD of ~ 200 m, which suggests that convection occurs across a limited depth. It is very possible that dense waters are formed in shallow coastal areas and that the density anomaly is then propagated on the seafloor by the bottom boundary layer scheme of NEMO. But I am not sure the latter is convection as people generally think. Alternatively, it would be interesting to output daily variables to see if the MLD monthly max of ~ 200 m is the result of one or a couple of deeper convective episodes intertwined with the absence of convection. In this case, the term "deep-water formation" would probably be more justified.

This is a very valid point. We agree that we don't simulate open-ocean convection down to the seafloor in the South Atlantic and admit that this needs to be highlighted. We argue that there were potentially two relevant mechanisms. Convection (< 200 m) of warm and saline waters in the subtropical gyre of the South Atlantic and formation of very saline and dense waters around the shelf

regions and particularly in the Angola Basin. These were the densest locally-formed water masses and might have entered the Cape-Argentine Basin as overflow waters, following down the topography of the Walvis Ridge (Fig. 10). We think it is important to discuss these two mechanisms here and added a clarification that the bottom waters were not formed by convective cells (in contrast to the high-latitude Pacific Ocean), but rather by sluggish and dense overflow waters from the northernmost shelf regions in the revised manuscript (l. 467-469).

References:

Greve, P., Orlowsky, B., Mueller, B. *et al.* Global assessment of trends in wetting and drying over land. *Nature Geosci* **7**, 716–721 (2014). https://doi.org/10.1038/ngeo2247

Huber, B. T., MacLeod, K. G., Watkins, D. K., and Coffin, M. F.: The rise and fall of the Cretaceous Hot Greenhouse climate, Global and
Planetary Change, 167, 1–23 (2018), https://doi.org/10.1016/j.gloplacha.2018.04.004

**Response to Reviewer #2**

We thank reviewer #2 for their positive evaluation of our work and for providing very helpful suggestions for further improvement. Below we provide a point-by-point reply to all comments raised. Reviewer comments are in black, our response is in red font. Line and figure numbers in our responses are based on the revised manuscript.

In their manuscript *Controls on Early Cretaceous South Atlantic Ocean circulation and carbon burial – a climate model-proxy synthesis'* by S. Steinig et al. used an ensemble of 36 coupled ocean-atmosphere simulations to evaluate South Atlantic Ocean circulation during the early Cretaceous. They setup the simulations in order to account for the uncertainty in basin configuration, regional seaways (Drake passage and Walvis ridge) and pCO2. To do so they account for all possible combination of forcing. They then carefully analyzed the contribution of each factor on oceanographic changes in the Southern Atlantic basin. In the last part of the paper, they integrate simulated scenarios with eNd data and sedimentological information to discuss evolution of Southern Ocean oceanography during the Barremian-Aptian.They notably highlight that paleogeography changes, happening at higher temporal resolution than those commonly investigated with paleo-climate modeling might have very strong impact locally, which would be reflected in the proxy.

I find the methodology very interesting. Those ensemble simulations, that are now accessible with classic ESM/GCM thanks to improved model performance seems to be one of the steps forward for deep time paleo-climate modeling, though they are not often used yet. The choice of boundary conditions and forcing are well justified in the text and relevant with regard to existing literature. The manuscript is overall super easy to read, and I appreciate the care that have been given to figures. I really enjoyed reading this paper and find remarkable that I am almost left with nothing to say. So congrats!

Thank you for the thorough summary and the positive feedback regarding the study design.

I only have minor comments:

- Authors analyze the dynamics of Southern Atlantic as isolated from the global circulation. I wonder whether the boundary conditions they implemented have also an impact of global circulation and whether global changes could affect the regional signal. For example, do the water masses transported through Drake Passage and entering the Southern Atlantic always have the same characteristics? Or does changes in source water characteristics might also impact the paleoceanography of the Southern Atlantic basin?

This has also been recommended by reviewer #1 and we agree with the usefulness of a description of the mean state and sensitivity of the global ocean circulation. In our simulations, we do find an important influence of the Drake Passage depth on the large-scale ocean circulation and the associated meridional heat transport. For this, we extended the simulations of three ensemble members with three different Drake Passage depths for a further 1500-2000 model years each to analyse changes in the global equilibrium circulation. We show these results in the new Fig. 3 and Section 3.1 ("Global ocean circulation"; I. 176-190) in the revised manuscript.

Figure 1 – 2 : It could be useful to have the location of basins indicated on the maps as well so it's easier to visualize. While it's not essential for understanding what the authors are writing it would help reader not always familiar with local geology structures to better understand.

We added basin labels to Figure 1 to have a common reference for all basin and gateway names used throughout the study.

Figure 1 : Color distinction between sites 249 and 693 is not strong enough.

Agreed. The colour for site 249 has been changed in the revised Fig. 1.

Figure 2 : would be good to have a plot showing global time series for the sensitivity as well.

Yes, we added this plot as the new Fig. S2 to the Supplementary Material and present the results in Section 2.3 (l. 168-172). This plot now also includes the three extended spin-up integrations to test for deep ocean equilibrium mentioned above. It shows that the existing ensemble members are already near equilibrium – even in the deep ocean – as the temperature drift over the additional 1500-2000 model years of all extended simulations is below 0.3 °C at all levels.

Figure 4c-d : This one is a very minor suggestion but changing the color scale for c) and d) to a scale that have various shade of the same color (like white – light blue to dark blue for example) would help better emphasize where temperature/salinity are highly variable and where variability is small.

We agree that this would help to highlight areas of small/high variability and updated the colour scales for panel (c) and (d) in the revised Fig. 5 accordingly.

L.185 – Note that the maxima in salinity and temperature are not strictly located at the same depth: one looks to be in the subsurface while the other is at the surface in the figure. Also *"the southern South Atlantic and the Angola Basin both show pronounced"* – do the authors mean Cape-Argentine basin instead of southern Atlantic ? Because this would make more sense in the sentence?

Yes, we will change this to "Cape-Argentine Basin" and remove the reference to a subsurface temperature maximum. The part (l. 208-209) will now read:
*"As a result of the salinity-driven intermediate water formation, the Cape-Argentine and the Angola basins both show a pronounced subsurface maximum in salinity at depths between 200-600 m (Fig. 5b)."*

L. 197-198 – authors should also refer to figure 5c in this sentence.

We added this figure reference as suggested (l. 220-221).

L.275 – *"somewhat surprising result is that the doubling of atmospheric CO2 does not only increase evaporation over the ocean but also reduces local precipitation, contradicting a simple dry gets drier, wet gets wetter paradigm."* This sounds that it is a general response while looking at figure S6 at first order region of E-P > (<) 0 have increased (decreased) E-P with CO2 doubling no matter if this is driven more by changes in precipitation or evaporation. I suggest that the authors reformulate this sentence.

Yes, the sentence indeed intended to highlight that the northern Angola Basin behaves differently than the zonal mean, as it switches from a precipitation to an evaporation-dominated region between the 600 ppm and 1200 ppm simulations. We will split this sentence into two to make this contrast clearer. The part (l. 302-306) will now read:
*"A somewhat surprising result is that the doubling of atmospheric CO2 does not only increase evaporation over the ocean but also reduces local precipitation over the most northern grid points of the Angola Basin, resulting in a switch from a previously precipitation to now evaporation-dominated region in response to a CO2 doubling. This is different compared to the dry gets drier, wet gets wetter pattern simulated for most of the ocean surface (Supplementary Fig. S7)."*

L337 - *Anagola Basin* --> Angola Basin ?

Changed. (l. 371)

L385 - Walvis Ridge Overflow Water – can you plot that on you maps/cross section to improve readability?

The WROW label has been added to Fig. 13c in the revised manuscript.

Figure S1 : to a horizontal grid of 0.5◦ x 0.5◦ horizontal grid  --> to a horizontal grid of 0.5◦ x 0.5◦ . Please also specified that this is the case for Perez-Diaz & Eagles, 2017 but that KCM plots are at model resolution.

The sentence in the caption of Fig. S1 has been changed to:
*"Only the minimum and maximum reported water depths are shown for each stage. KCM bathymetries are shown on the native model grid, while the reconstructions have been re-gridded to a 0.5° x 0.5° horizontal grid.*

Figure S6 vs Figure 10. Please chose between P-E or E+P notation.

We chose a different notation in Fig. S7 because the evaporation fluxes are shown with a negative sign in panels (d-e). We made sure that the E-P notation is consistent in the main manuscript for the new Figures 4 and 11 and we added a note to highlight this difference in the caption of Supplementary Fig. S7.

[Figure]

**Supplementary Figure S1:** South Atlantic meridional overturning circulation variability. Results are shown for a 100-year window of a single ensemble member (stage 3, 1400 m Drake Passage, 1200m Walvis Ridge, 1200ppm CO2) for illustration purpose. Panels show the (a) time mean, (b) interannual standard deviation and (c) time series of the maximum overturning of the domain. All values are in Sv.

[Figure]

**Supplementary Figure S2:** Annual cycle of South Atlantic mixed layer depth. Results are shown for a single ensemble member (stage 3, 1400 m Drake Passage, 1200m Walvis Ridge, 1200ppm CO2) for illustration purpose. Values are in m.